# Analysis of Three-Phase Inverter Parallel Operation with Network-Based Control Having Strong Robustness and Wide Time-Scale Compatibility in Droop-Controlled AC Microgrid

**Yao Zhang** [1,*] **, Fan Zhang** [1] **, Yu Quan** [1] **and Pengfei Zhang** [2]

[1] School of Automation, Hangzhou Dianzi University, Hangzhou 310018, China; zhangfan@hdu.edu.cn (F.Z.); quanyu@hdu.edu.cn (Y.Q.)
[2] State Grid Zhejiang Electric Power Company, LTD Hangzhou Power Supply Company, Hangzhou 310020, China; zpf365@163.com
* Correspondence: yaozhang@hdu.edu.cn; Tel.: +86-571-8771-3513

**Abstract:** The system performances can be potentially enhanced for three-phase inverter parallel operation in droop-controlled AC microgrid by using network-based control, which also benefits for the extension of other control strategies in microgrids (MGs). It is highlighted that some negative factors such as network-induced time-delays and data dropouts would possibly degrade the system operation. In this paper, the comprehensive analysis of network-based control strategy with strong robustness and wide time-scale compatibility is investigated in islanded mode of an AC microgrid with paralleled inverters. The theoretical evaluation towards time-delay and data dropouts is made and it is verified that its good power-sharing can be obtained under unsatisfactory communication conditions. It has been observed that the time-scale of network-based control can also be designed from several microseconds to milliseconds. Based on this idea, the communication integration of different layers of MGs in hierarchical structure would be realistic. Experimental results have verified the effectiveness of the network-based control strategy and analytical method.

**Keywords:** inverter parallel; network-based control; data dropout; time-delay; droop-controlled microgrids; time-scale compatibility

## 1. Introduction

During the last decades, the power flow pattern has changed with the increasing penetration of distributed energy from large wind farms, geothermal power plants, and solar photovoltaic power stations [1]. Nowadays, the prodigious growth rate of plug-in hybrid vehicles (PHEVs) is accelerating this power system change, which means conventional generation is gradually being replaced and distributed generation (DG) is playing an important role in the future smart grids [2,3]. The concept of microgrids (MGs) emerged at the beginning of this century and is capable of operating in the grid-connected mode and islanded mode and handling the transitions between these two modes [4,5].

In AC microgrids (MGs), renewable energy generation is increasingly integrated into the grid through power electronic converters such as DC-AC inverters, which are always operated in parallel [6]. When an MG is working in islanded mode, these inverters are always employed by droop control to realize satisfactory load-sharing performance and balance the demand of local loads to ensure system stability [5,7]. Meanwhile, the system regulation including frequency deviation and voltage droop, will be implemented in different time-scales [8]. The control relationships of these MGs have three main categories, i.e., the master-slave control, peer-to-peer control, and hierarchical control. As for the

master-slave control strategy, one inverter is treated as the master and the rest are slaves. The system safety and reliability are preponderantly dependent upon the master. The peer-to-peer control indicates the inverters have equal status, which avoids the communication links and is capable of plug-and-play operation when the micro-sources join or quit the MG whenever possible [5]. However, these two strategies are individually used in islanded mode or grid-connected mode, and the hierarchical control strategy is proposed to combine both of these two modes and solve their switching problems.

Although the control strategies are different, they may use communication links among the inverter controllers more or less [9–15], accordingly the network-based control was proposed. The introduction of network is mainly used to improve the performance and consolidate the system stability, as is the hierarchical structure. In addition, the power electronic converter in the MG has some features such as low inertia and fast response [16]. The power system built by the MG is quite different from conventional power systems, showing a wide range of time-scale properties. The hierarchical design of the MG can distinguish these time-scales and fulfill their functions of electrical control, power quality regulation, and economic operation control independently in their exclusive time-scale, which is helpful to realize function standardization and improve the intelligence and flexibility of the MG [5,17]. The primary layer in the hierarchical structure containing most power electronic converters seldom uses network or only uses low-bandwidth network, then the droop control is used to solve the load-sharing issues among the paralleled micro-sources [18]. The secondary control layer is employed to eliminate the deviation of output voltage and frequency existing in the primary layer, the structure is possibly centralized or decentralized and implemented partly by network configuration. However, the droop control employed in primary control has several disadvantages that limit its application, such as relatively inaccurate load-sharing, slow instantaneous response, frequency and amplitude deviation, and strong susceptibility to converter output impedance, etc. Especially when lots of different types of loads such as PHEVs plunge into the MG, it is hard to hand these disturbances from the sources to the loads safely and keep system permanent stable.

Many advanced methods based on droop control had been reported for inverter parallel operation [19–22]. For example, some strategies were presented such as Complex Line Impedance-Based Droop Method [23], Angle Droop Control [24], Voltage-Based Droop Control [25], and Adaptive Virtual Impedance Control [26]. These methods can partially solve the problems of link impedance unbalance, non-linear load-sharing, voltage distortion, and so on. However, they were trying to avoid the communication usage, at the expense of adding high computational complexity and extra analytical prerequisites. More importantly, they always achieved slightly inferior performance than the methods with interconnections. Under this circumstance, network-based control strategies were proven to be effective for the performance optimization of MGs [18]. The interconnections can be designed from the physical line to network with a wide range of bandwidth. In this way, long-term stability, higher reliability, and superior load-sharing performances in different time-scales can be obtained via network. Moreover, the network in the primary layer could be integrated with the secondary layer if their communication time-scales are matched. However, some uncertain factors related to communication conditions such as time-delays or data dropouts would influence the system in the different ways [27]. How to evaluate the impact of these factors and define the stability region of the system having these delays and data dropouts is very important.

In this paper, a comprehensive analysis of network-based control strategy used in the islanded mode of an AC microgrid is investigated, the model is simplified as a paralleled three-phase inverters system and it is a continuation of [28]. In general, the merits network-based control can bring (1) simple and convenient implementation, (2) only one communication line needed to fulfill the bi-directional data flow, (3) better power-sharing performance than the traditional droop method, (4) anti-interference ability to restrain disturbances such as network-induced negative factors, (5) no extra control loop with a complicated algorithm, such as a virtual impedance loop, to supplement. Moreover, more advantages of the method including strong robustness and good time-scale compatibility can be achieved through analysis. Theoretical and experimental methods were utilized to verify these

characteristics. The proposed analysis method and the obtained advantages of the network-based control strategy are worthy of studying because (1) the transmission speed and allowable capacity of the network to keep the system stable are in a wide range, and the network-based control can be compatible with both fast and slow loops. Moreover, the communication design is flexible and can be integrated with a power electronic converter and it is beneficial to build a new MG structure. (2) In a wide time-scale range, the system stability and performance may be immune to the time-delays and data dropouts. The high reliability it creates is helpful to maximize network utilization in power electronics. It is necessary to mention that many different communication infrastructures, such as Ethernet, worldwide interoperability for microwave access (WiMAX), and wireless fidelity (Wi-Fi), can be the good alternatives in network-based control in MGs [5]. With the development of highly reliable and ultrafast networks, the impact of time-delays and data-dropouts on MG network-based control would possibly decrease.

This paper is organized as follows. Section 2 provides the envisaged architecture of the network-based control system with paralleled three-phase inverters and addresses its potential role. Section 3 presents the network-based control methods. Section 4 builds the mathematical model featuring system sensitivity towards time-delay and data dropout, provides the analysis methods to specify their system stability. Experimental results of the proposed analysis are given in Section 5. The supplementations of some issues including different rated power, time-delay level, and abnormal communication conditions are described in Section 6. Section 7 concludes the main contribution of this paper.

## 2. Architecture of Network-Based Control in MGs

The operation time of the primary layer, the secondary, and tertiary level can be microseconds, seconds, and minutes. Hierarchical design is aiming for independent task-finishing for these different time-scale layers. We suppose the primary layer control is working without communication links, then droop control and its modified methods are available. As discussed above, these kinds of methods have their inherent disadvantages. For long-term stability and reliability, a network-based method could be possibly required to support the primary layer. If the network has powerful data-scheduling capability and the same time-scale level, the communication could be integrated together between the primary and secondary layer, which is cost-saving and increases utilization. That is the envisaged idea proposed in this paper, architecture of which is shown in Figure 1 (the tertiary layer is not considered here). There are $N$ inverters in parallel micro-sources of MGs to support electricity of the load, where power droop control is employed to guarantee the basic load-sharing performance and system stability. The network-based control plays a helpful role in enhancing high-level reliability and stability.

In primary control, there are at least two kinds of loops, the inner fast current/voltage loop and the outer slow power-droop loop. Sometimes, the power loop features an open loop unless virtual impedance or other strategy is added. In this paper, a novel idea related to the control loop is proposed, where the third control loop is designed to help the system in data-interaction via a network, in which the loop data flow is marked as a red arrow line and shown in Figure 1. As is well-known, the network transmission speed is adjustable and thereby the time-scale of the extra loop could be flexibly designed. That means it could be slower than or even comparative to the inner loop and power loop. Thereby, how to design the third loop and how to evaluate the impact of network-induced time-delays or data dropouts during transmission are very important.

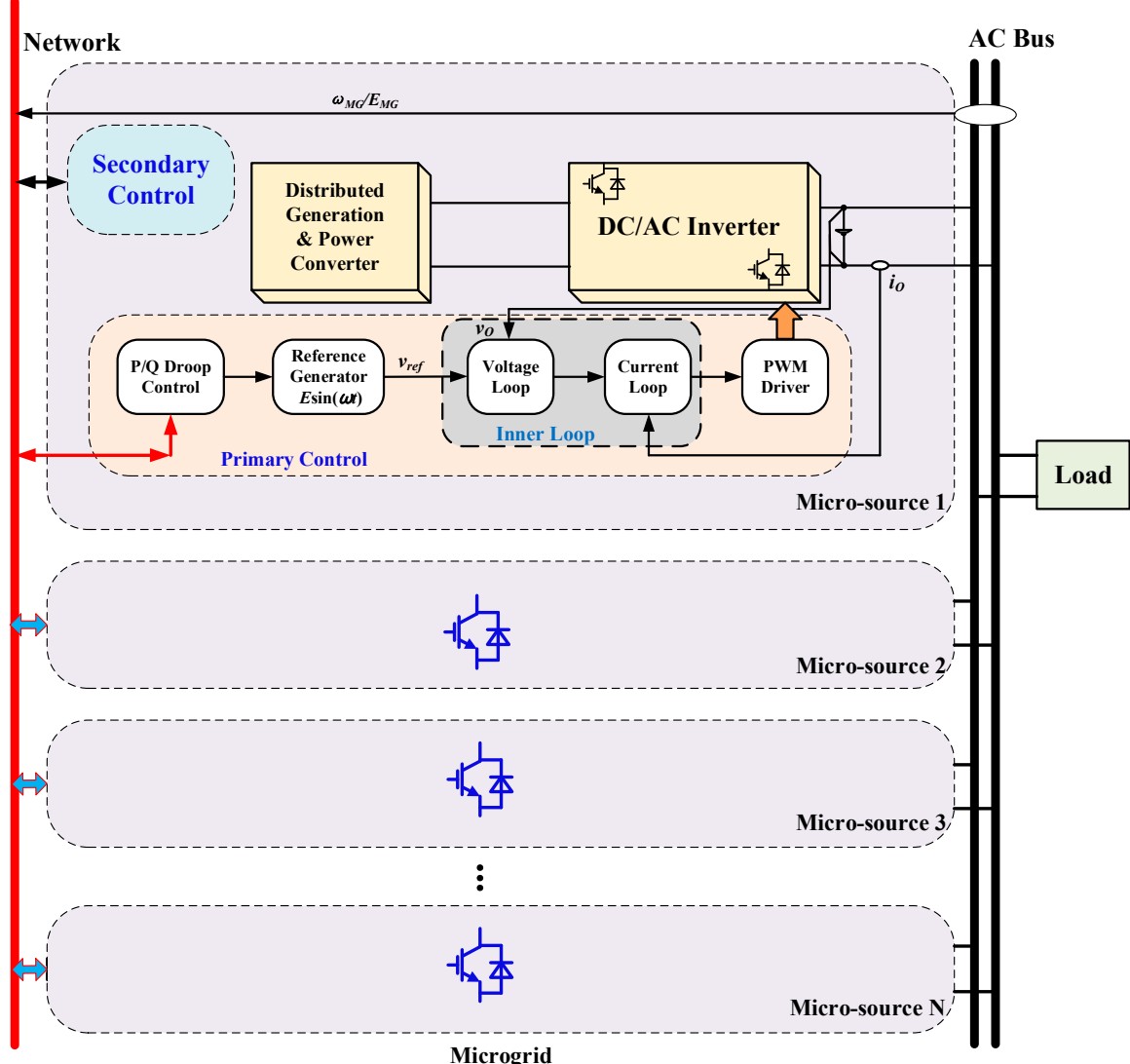

**Figure 1.** Envisaged architecture of parallel micro-sources with network-based control based on hierarchical structure.

## 3. Network-Based Control Strategy of Load-Sharing

### 3.1. Traditional P/Q Droop Control

In order to analyze the network-based control, the equivalent circuit of micro-source paralleled to an AC bus is outlined as shown in Figure 2. To facilitate the analysis, distributed generation and its converter for each MG can be simplified as the ideal DC source, so that the circuit is equivalent to the system with three-phase inverters in parallel. As shown in Figure 2, the sum of output impedance and link impedance of the single inverter is $\dot{Z}_n = Z_{on}\angle\theta_{on} + Z_{cn}\angle\theta_{cn} = Z_n\angle\theta_n$. $E_n$, $\varphi_n$, and $V$ are the amplitudes and power angle of the inverter output voltage and the common bus voltage, $S_n$, $P_n$, and $Q_n$ are the complex power, active power, and reactive power in output load provided by the $n^{\text{th}}$ inverter, respectively.

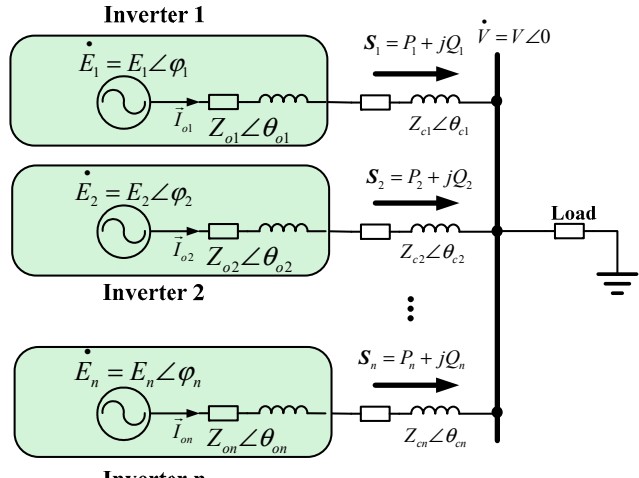

**Figure 2.** Equivalent circuit of inverters paralleled to an AC bus.

The complex power dawn to the bus can be expressed as

$$S_n = E_n\angle\varphi_n \cdot \dot{I}_n^* = P_n + jQ_n, \tag{1}$$

where $P_n$ and $Q_n$ are the active and reactive power, respectively,

$$\begin{aligned} P_n &= \frac{V}{Z_n}[(E_n\cos\varphi_n - V)\cos\theta_n + E_n\sin\theta_n\sin\varphi_n] \\ Q_n &= \frac{V}{Z_n}[(E_n\cos\varphi_n - V)\sin\theta_n - E_n\cos\theta_n\sin\varphi_n] \end{aligned}. \tag{2}$$

If they have the same frequency of output voltage and small power phase, then

$$\begin{aligned} P_n &= \frac{V}{Z_n}[(E_n - V)\cos\theta_n + E_n\sin\theta_n\varphi_n] \\ Q_n &= \frac{V}{Z_n}[(E_n - V)\sin\theta_n - E_n\cos\theta_n\varphi_n] \end{aligned}. \tag{3}$$

Based on Equation (3), the relationship between power and power angle is derived and shown in Table 1. When system impedance is inductive, the active power is mostly dependent on the power angle and the reactive power is affected by the amplitude. Then the traditional P/Q droop control is obtained, the active and reactive power P/Q are regulated by controlling the amplitude and power angle of output voltage [29].

$$\begin{aligned} E &= E_{ref} - k_{qe}Q \\ \omega &= \omega_{ref} - k_{p\omega}P \end{aligned}. \tag{4}$$

**Table 1.** The equation of active power $P$ and reactive power $Q$ with varied system impedances.

| System Impedance | $\theta = 0°$ | $\theta = 90°$ | $\theta$ |
|---|---|---|---|
| $P$ | $P = \frac{(E-V)V}{Z}$ | $P = \frac{EV}{Z}\varphi$ | $P_n = \frac{V}{Z_n}[(E_n - V)\cos\theta_n + E_n\sin\theta_n\varphi_n]$ |
| $Q$ | $Q = \frac{-EV}{Z}\varphi$ | $Q = \frac{(E-V)V}{Z}$ | $Q_n = \frac{V}{Z_n}[(E_n - V)\sin\theta_n - E_n\cos\theta_n\varphi_n]$ |

The other scenarios with different system output impedances can be considered in the same manner as shown in Table 1. It is clearly demonstrated that there exists strong coupling between P/Q and impedance angle.

*3.2. Output Impedance of Three-Phase Inverter in dq0 Coordinate*

The main circuit of the studied three-phase three-line DC-AC inverter is shown in Figure 3, where the load terminal is designed as a three-phase symmetrical delta connection. To make the analysis

simplified, input is the ideal DC source $V_{dc}$. SPWM (Sinusoidal Pulse Width Modulation) is the modulation method and there are two control loops inside, the current and voltage loop. The former is aiming at the enhancement of system dynamic feature and therefore the gain crossover frequency is large. The latter has small gain crossover frequency and can be used to improve the stability performance. The control block diagram of inverter in *dq*0 coordinate is shown in Figure 4, where control output of voltage loop, $i_{dref}$ and $i_{qref}$, are the reference values of the current loop in dq0 coordinate. $K_{vp}$, $K_{vi}$, $K_{ip}$, and $K_{ii}$ are the control coefficients of the inner loop.

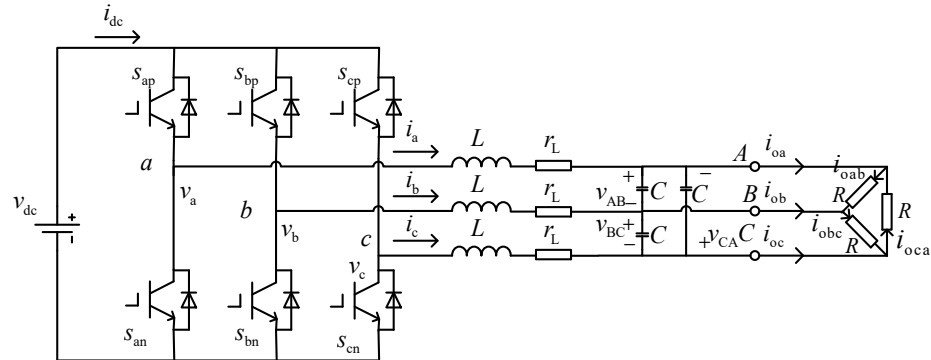

**Figure 3.** Schematic diagram of three-phase inverter.

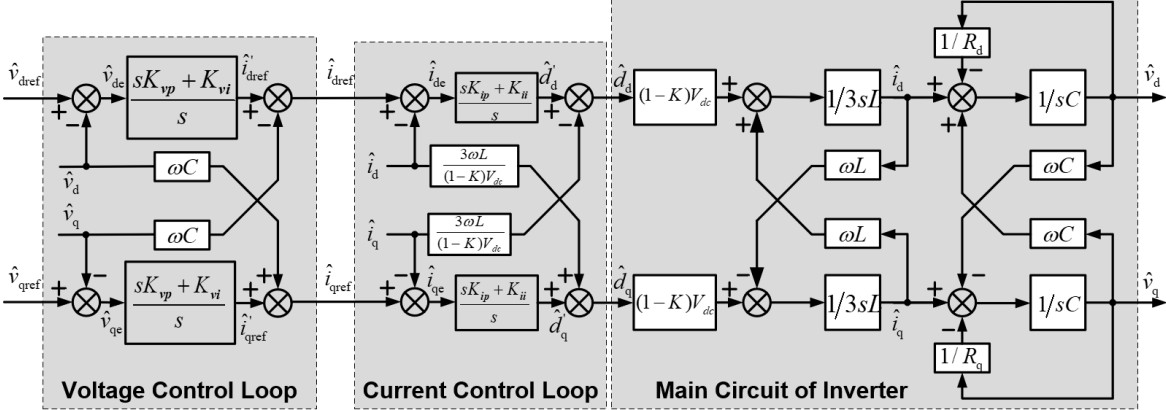

**Figure 4.** Control block diagram of inverter inner control loop in dq0 coordinate (by using small-signal model).

The small-signal model shown in Figure 4 in dq0 coordinate can be described as mathematical model,

$$\begin{cases} \hat{v}_d(s) = G_{od}(s)\hat{v}_{dref}(s) + G_{qd}(s)\hat{v}_q(s) - Z_{od}(s)\hat{i}_{od}(s) \\ \hat{v}_q(s) = G_{oq}(s)\hat{v}_{qref}(s) - G_{dq}(s)\hat{v}_d(s) - Z_{oq}(s)\hat{i}_{oq}(s) \end{cases}. \tag{5}$$

In the dq0 coordinate system, the ac variables of power frequency in the original *abc* coordinate system become the dc variable, i.e., $G_{od}(0) = G_{oq}(0) = 1$, $G_{qd}(0) = G_{dq}(0) = 0$, $Z_{od}(0) = Z_{oq}(0) = 0$. Corresponding to the parameters of experimental platform, $V_{dc}$ = 250 V, inductance $L$ = 3.4 mH, voltage reference $V_{dref}$ = 150 V, $V_{qref}$ = 0 V. Amplitude modulation ratio $m = V_{dref}/(0.866 \times V_{dc})$ = 0.7, $K = 8f_c t_d/m\pi$ = 0.15. In addition, switching frequency $f_c$ = 20$kHz$, dead-time is $t_d$ = 2$\mu s$, load is set as $R$ = 70$\Omega$, filter capacitor is $C$ = 2.2$\mu F$.

As shown in Figure 5, the amplitude of output impedance $Z_{od}$ (s) tends to zero and phase angle is approaching 90° in the low frequency. In other words, the output impedance $Z_{od}$ (s) of the fundamental component is basically inductive in the dq0 coordinate system in this case.

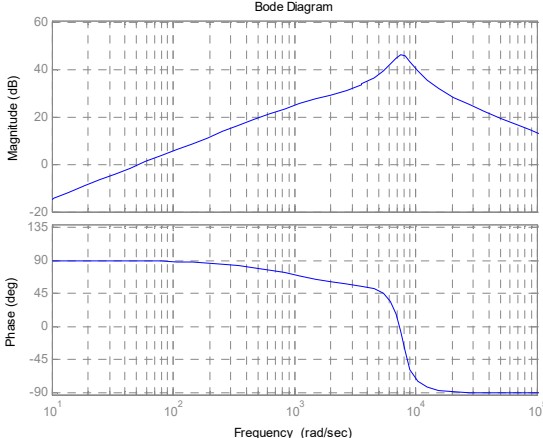

**Figure 5.** Output impedance of inverter $Z_{od}(s)$ in dq0 coordinate.

### 3.3. Analysis of Parallel Operation in dq0 Coordinate

Due to the inductive feature of output impedance $Z_{od}(s)$, the three-phase inverter parallel operation has the basic load-sharing control strategy, which can be deduced based on Equation (4) and shown in the following

$$\begin{cases} \omega_n = \omega_{ref} - k_{p\omega n}P_n \\ v_{drefn} = V_{dref} - k_{qVn}Q_n \\ v_{qrefn} = V_{qref} = 0 \end{cases}, \tag{6}$$

where $\omega_{ref}$ and $V_{dref}$ are reference frequency and voltage amplitude of the $n^{\text{th}}$ inverter with no-load in $d$ axle whereas reference voltage of $q$ axle is zero. $P_n$ and $Q_n$ are the active and reactive power, $k_{p\omega m}$ and $k_{qVn}$ are the droop coefficients. The modified reference values $\omega_n$ and $v_{drefn}$ are calculated and used as the references of the inner voltage loop for the $n^{\text{th}}$ inverter. The control diagram of power droop strategy is shown in Figure 6, where the output becomes the input of voltage and current control. Instantaneous power $p$ and $q$ are calculated by the output voltage and current variables $v_d$, $v_q$, $i_{od}$, and $i_{oq}$ in dq0 coordinate. The filtered power $P$ and $Q$ are substituted into droop Equation (6) to generate the updated angular frequency $\omega$ and voltage amplitude $v_{dref}$.

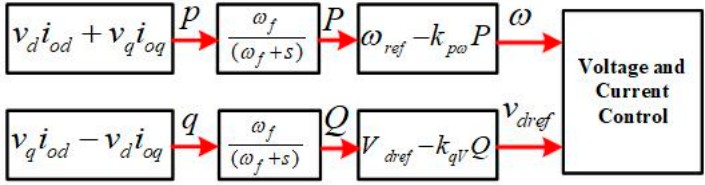

**Figure 6.** Control block diagram of the power droop control for an inverter.

The control power $P_n$ and $Q_n$ of the $n^{th}$ inverter are obtained by passing the instantaneous power $p_n$ and $q_n$ through the low-pass filter. To be simpler, the first-order filter is used and equivalent as follows:

$$\begin{cases} P_n(s) = \frac{\omega_f}{s+\omega_f}p_n(s) \\ Q_n(s) = \frac{\omega_f}{s+\omega_f}q_n(s) \end{cases}, \tag{7}$$

where $\omega_f$ is the cut-off frequency of the low-pass filter. In addition, the instantaneous power can be obtained from sampled voltage and current in $dq$ axes,

$$\begin{cases} p_n = v_{dn}i_{odn} + v_{qn}i_{oqn} \\ q_n = v_{qn}i_{odn} - v_{dn}i_{oqn} \end{cases}. \tag{8}$$

According to the power droop method, the active power and reactive power output of each inverter at steady state satisfy the following equation:

$$\begin{cases} k_{p\omega 1}P_1 = k_{p\omega 2}P_2 = \cdots = k_{p\omega n}P_n \\ (k_{qV1}V_{d1} + Z_1)Q_1 = (k_{qV2}V_{d2} + Z_2)Q_2 = \cdots = (k_{qVn}V_{dn} + Z_n)Q_n \end{cases}, \tag{9}$$

where $Z_n(s) = Z_{odn}(s) + Z_{cn}(s)$. In dq0 coordinates, for the fundamental component, there is $Z_n(0) = r_n$.

Generally speaking, the distance between the inverter module and the load is relatively long, then $k_{qVn}V_{dn} \gg r_n$. In addition, $V_{dn} \approx V_{dref}$, Equation (9) can be simplified in the following:

$$\begin{cases} k_{p\omega 1}P_1 = k_{p\omega 2}P_2 = \cdots = k_{p\omega n}P_n \\ k_{qV1}Q_1 = k_{qV2}Q_2 = \cdots = k_{qVn}Q_n \end{cases}. \tag{10}$$

It is obvious that the change of $k_{p\omega n}$ and $k_{qVn}$ will lead to the diversification of the power ratio among the paralleled inverters.

### 3.4. Network-Based Control of Power-Sharing for Paralleled Three-Phase Inverters

The difference between network-based control and the methods with wires is obvious. The network-based control can flexibly regulate the time-scale of control loop and manage the data flow via tuning transmission schedule, meanwhile the negative impact of unusual conditions such as time-delay, data dropout or network failure on system stability needs to be minimized, which is solvable in most cases. As for the wire-linking methods, there are no such regulating effects of interconnection. As a result, it is very likely to lead to the failure of the system operation if the interconnection is broken. As discussed in Section 2, the time-scale of extra network-based control loop could be compatible with other fast/slow loops, then it could achieve the communication integration with secondary control layer as illustrated in Figure 1. Thus, the simple architecture of the network-based control of inverter parallel operation is shown in Figure 7, where network is an abundant resource to control information, and the desirable medium to fulfill the network-based control loop. Each controller of the three-phase inverter sends their control variable to the network following the blue arrow direction and receives the reference information from the network with the guidance of red arrow. The extra closed-loop of network-based control is naturally formed. It is clearly indicated that control decision is generated from the network, the network is capable of calculating the control reference based on the information uploaded by all of the three-phase inverters.

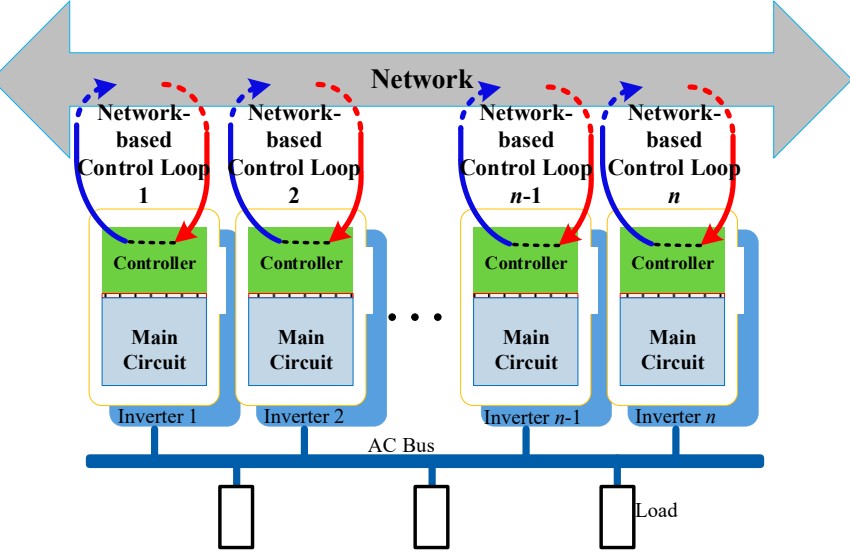

**Figure 7.** Architecture of network-based control with multi three-phase inverter parallel operation.

The other role of the network, the inherent advantage as well, is the flexible allocation of data resources. It needs to be mentioned that the network-based control loops are not isolated since they are connected in the network. Therefore, the coupling effect of these loops are certainly concluded into the control strategy. The network-based control is built based on power droop control as shown in Equation (6). $\omega_n$, $v_{drefn}$, and $v_{qrefn}$ are generated to produce the voltage reference for voltage/current inner control loops, $k_{p\omega n}$ and $k_{qVn}$ are usually set as constant coefficients. In addition, $P_n$ and $Q_n$ are the key variables to affect the system stability and control precision. Inspired by the idea of [18], the outer power control loop based on $P_n$ and $Q_n$ could be modified as a network-based strategy. The network is one tool for collecting, analyzing, and dispatching the data, as a result all inverters become potential information sources and participators. According to the different direction of data flow, two modes of network-based control are presented, one is a master-slave mode and the other is an autonomous mode.

The schematic diagram of the two modes in network-based control is addressed in Figure 8. The master-slave mode has the status of master and slave relationship in a literal meaning. The master inverter sends its own power information $P_{master}/Q_{master}$ via the network (shown in Figure 8a), whereas the other inverters receive $P_{master}/Q_{master}$ through the network to modify their outside and inner control loops. It is obvious that the selection and operation state of the master inverter are the ultimate determinant and restriction factor of system performance because the failure of the master inverter will directly lead to system collapse. Compared with the master-slave mode, the peer-to-peer relationship, which is defined as autonomous mode shown in Figure 8b, seems to be a good alternative. It is observed that the autonomous mode makes each inverter have the bidirectional data flow via network, whereas there is only unidirectional flow for each inverter in master-slave mode. In a sense, bidirectional flow could be more effective and extensive in information fusion among the inverters.

In view of the characteristics of the distributed three-phase inverter system and in order to make the system have better fault tolerance and scalability, the autonomous mode is adopted in this paper. There is no any particular order of priorities in this inverter system, and the power information is transmitted among the inverters. Taking three inverters as an example ($i$, $j$, $k$ = 1,2,3), the inverter $i$ firstly calculates its output active power $P_{CALi}$ and reactive power $Q_{CALi}$, and then transfers them to the other two modules via network. At the same time, it receives the power information $P_{COMj}/Q_{COMj}$ and $P_{COMk}/Q_{COMk}$ from the other two modules ($j$, $k$), and incorporates them into the droop control equation through the weighted mode. Therefore, the governing equation of power weighted average distribution in parallel systems of three inverters is shown as follows:

$$\begin{cases} \omega_1 = \omega_{ref} - k_{p\omega}[(1 - m_1 - m_2)P_{CAL1} + m_1 P_{COM2} + m_2 P_{COM3}] \\ v_{dref1} = V_{dref} - k_{qV}[(1 - n_1 - n_2)Q_{CAL1} + n_1 Q_{COM2} + n_2 Q_{COM3}] \end{cases}, \tag{11}$$

$$\begin{cases} \omega_2 = \omega_{ref} - k_{p\omega}[(1 - m_1 - m_2)P_{CAL2} + m_1 P_{COM1} + m_2 P_{COM3}] \\ v_{dref2} = V_{dref} - k_{qV}[(1 - n_1 - n_2)Q_{CAL2} + n_1 Q_{COM1} + n_2 Q_{COM3}] \end{cases}, \tag{12}$$

$$\begin{cases} \omega_3 = \omega_{ref} - k_{p\omega}[(1 - m_1 - m_2)P_{CAL3} + m_1 P_{COM1} + m_2 P_{COM2}] \\ v_{dref3} = V_{dref} - k_{qV}[(1 - n_1 - n_2)Q_{CAL3} + n_1 Q_{COM1} + n_2 Q_{COM2}] \end{cases}, \tag{13}$$

where $m_1 \leq 1$, $m_2 \leq 1$, $n_1 \leq 1$, $n_2 \leq 1$ are the power weighting coefficients of network data, respectively. In this case, the power droop coefficients, $k_{p\omega}$ and $k_{qV}$, of three inverters are the same. $P_{CALn}/Q_{CALn}$ are the real-time, self-calculating power, whereas $P_{COMn}/Q_{COMn}$ are the network-transmitted power. In this sense, the novelty of the network-based control is that the power information can be exchanged and shared by each other via the network, the final active and reactive power control variables are formulated through the weighting algorithm, combining its real-time simultaneous power and the network power data. The network with the combination of CSMA+ (Carrier Sense Multiple Access) priority setting can handle the communication conflicts whenever possible. The CAN (Controller Area Network) bus featured via CSMA/CA (Carrier Sense Multiple Access with Collision Avoidance) link mode is a desired alternative in this network-based control implementation.

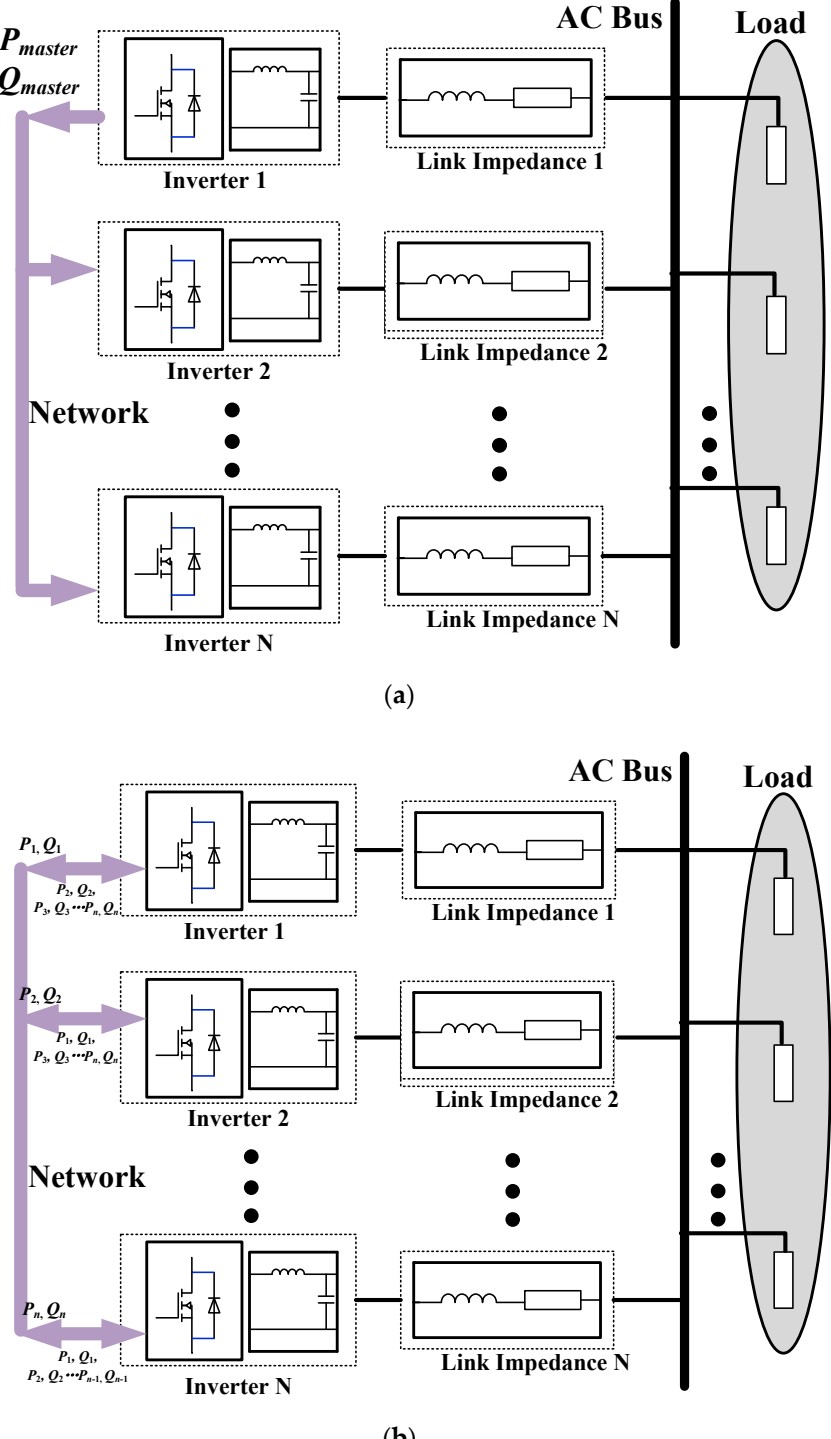

**Figure 8.** Schematic diagram of two modes in network-based control with inverter parallel operation.
(**a**) Master-slave mode, (**b**) autonomous mode.

The network-based control can be expanded to the application with *n* paralleled inverters, the control block diagram is shown in Figure 9. Compared with traditional droop control as shown in Figure 6, the power information of each inverter is dispatched to the network and then broadcast to other inverters, which helps to modify the traditional power droop control. In Figure 9, the power $P_j$ and $Q_j$ are obtained from real-time calculation, and the power control variables are generated by the combination functions with $P_j/Q_j$ and network data $P_{com,i}$ $(i \neq j)$.

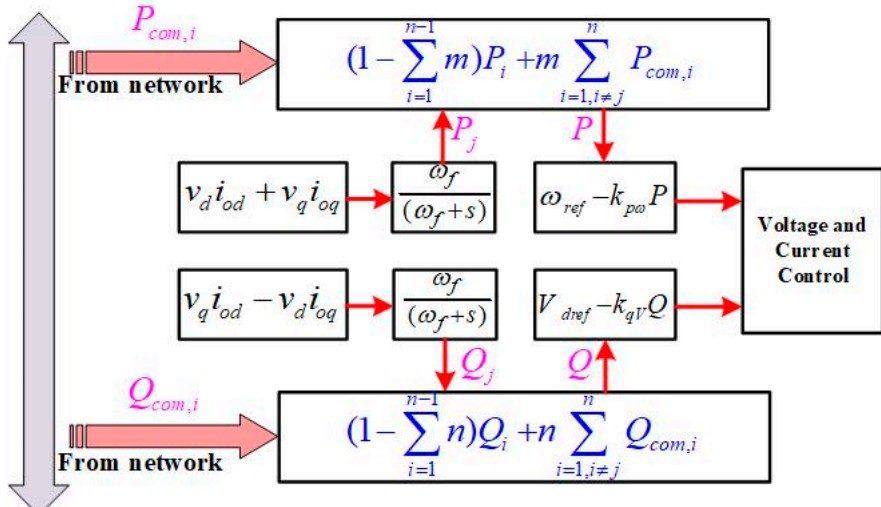

**Figure 9.** Control block diagram of the network-based control for the $j^{th}$ inverter.

## 4. Modeling Analysis on Network-Based Control Strategy

There is also a flipside that shows us the problems and challenges when we take advantage of the convenience of the communications. The negative aspects such as time-delay and data dropouts will be a risk to the network-based control system or degrade the performance [27]. Therefore, how to evaluate the overall performance of the network-based control strategy becomes a compromise assessment. In this section, the time-delay and data dropout are investigated in the system stability and performance.

### 4.1. Stability Analysis of Time-Delay

Suppose that the time-delay is bounded, and the time-delay of each sample $\tau_k$ is less than one sampling period $h$, the system state equation of this network-based control is addressed by

$$\begin{aligned}
\dot{x}(t) &= Ax(t) + Bu(t), t \in [kh + \tau_k, (k+1)h + \tau_{k+1}), \\
u(t^+) &= -Kx(t - \tau_k), t \in \{kh + \tau_k, k = 0, 1, 2, \ldots\}
\end{aligned} \tag{14}$$

A linearized discrete-time model is adopted to describe the network-based control with autonomous mode, and description of matrix $A$, $B$, state variable $x$ and input variable $u$ are formulated by Equations (A10), (A14), (A18), and (A19) in Appendix A.

Where $u(t^+)$ is piecewise continuous and only changes at $kh+\tau_k$. If we define the augmented state vector $z(kh) = \left[x^T(kh), x^T((k-1)h)\right]^T$, the system equation can be rewritten as

$$z((k+1)h) = \widetilde{\Phi}(k)z(kh), \tag{15}$$

where $\widetilde{\Phi}(k) = \begin{bmatrix} \Phi - \Gamma_0(\tau_k)K & \Gamma_1(\tau_k) \\ -K & 0 \end{bmatrix}$, and $\Phi = e^{Ah}$, $\Gamma_0(\tau_k) = \int_0^{h-\tau_k} e^{As}B ds$, $\Gamma_1(\tau_k) = \int_{h-\tau_k}^h e^{As}B ds$.

Based on standardized augmented and discrete-time system matrix with certainty mode of Equation (15), one of the system exponential stability theorems used in network-based control with bounded time-delay, is the Schur delay-dependent stability criteria. The essence is to observe whether or not $\widetilde{\Phi}(\tau)$ in Equation (15) is a stable Schur matrix. The upper limit value of time-delay while keeping the system stable can be used to analyze the stability sensitivity of time-delay. If the allowable limit is very low, that means the network-based strategy is highly sensitive to the time-delay, and vice versa. In this section, one method for evaluating the impact of network-induced time-delay is presented in

multi-inverter system with network utilization, although the results derived from method of Schur stability is possibly conservative.

To give a simpler example to analyze, the two-inverter parallel operation is put on the table for theoretical model analysis. Then the network-based control is addressed as

$$
\begin{cases}
\omega_1 = \omega_{ref} - k_{p\omega}[(1-m)P_{CAL1} + mP_{COM2}] \\
v_{dref1} = V_{dref} - k_{qV}[(1-n)Q_{CAL1} + nQ_{COM2}]
\end{cases},
\tag{16}
$$

$$
\begin{cases}
\omega_2 = \omega_{ref} - k_{p\omega}[(1-m)P_{CAL2} + mP_{COM1}] \\
v_{dref2} = V_{dref} - k_{qV}[(1-n)Q_{CAL2} + nQ_{COM1}]
\end{cases}.
\tag{17}
$$

Using the model given in Equation (14) and the parameters listed in Table 2, the allowable bounds of the time-delay with the variant control parameters *m* and *n*, which is reflected in feedback gain matrix *K* in Equation (A18), are grouped into one chart by means of the LMI toolbox of MATLAB. The results are shown in Figure 10, with a six color scale. Dark red shows the maximum allowable time-delay is 20 ms or even more. Pink red and bright red mean the value is slightly decreased and are in the range of [18,20] [16,18] ms. The allowable time-delay is 2 ms subtracted from 16 ms each time and is labeled as orange and yellow. Dark blue is finally used to exhibit the unstable condition. Therefore, it is easily obtained that in most areas with different *m/n* combination, the maximum allowable time-delay to keep the system stable τ is equal to or approaching to *h* (in this case *h* is set at 20 ms). There is only a very small surface covering dark blue to show the vulnerability towards time-delay, which is addressed at *m* > 0.75 and *n* > 0.86. In the area labelled by dark red, there is a high possibility of a larger allowable time-delay if the restriction of the maximum time-delay being less than one sampling period *h* is released. In a sense, the allowable time-delay belongs to the tens of milliseconds with a majority of control coefficient combinations. An extra explanation on time-delay and its robustness is presented in Section 6.

**Table 2.** Specification of the network-based control system model.

| Parameters | Symbols | Values | Parameters | Symbols | Values |
|---|---|---|---|---|---|
| Input dc voltage | $V_{dc}$ | 250 V | Rectifier bridge capacitance in parallel | $C_{non}$ | 2700 μF |
| Reference voltage amplitude | $V_{dref}+jV_{qref}$ | 155+j0 V | Rectifier bridge resistance in parallel | $R_{non}$ | 50 Ω |
| Fundamental frequency | $f$ | 50 Hz | Rectifier bridge series resistance | $R_S$ | 2.2 Ω |
| Switching frequency | $f_c$ | 20 kHz | Voltage loop P parameter | $K_{vp}$ | 0.01 |
| Dead-time | $t_d$ | 2.0 μs | Voltage loop I parameter | $K_{vi}$ | 54 |
| Filter inductance | $L$ | 3.4 mH | Current loop P parameter | $K_{ip}$ | 0.2 |
| Filter inductance ESR (Equivalent Series Resistance) | $r_L$ | 0.2 Ω | Current loop I parameter | $K_{ii}$ | 10 |
| Filter capacitor | $C$ | 2.2 μF | Frequency drop coefficient | $k_{p\omega}$ | 0.001 |
| Load resistance | $R$ | 35/47/70 Ω | Amplitude droop coefficient | $k_{qV}$ | 0.006 |
| Line resistance | $r$ | 0.01 Ω | Active power allocation coefficient | $m$ | 0.3 |
| Line induction | $X$ | 0.09 Ω | Reactive power allocation coefficient | $n$ | 0.4 |

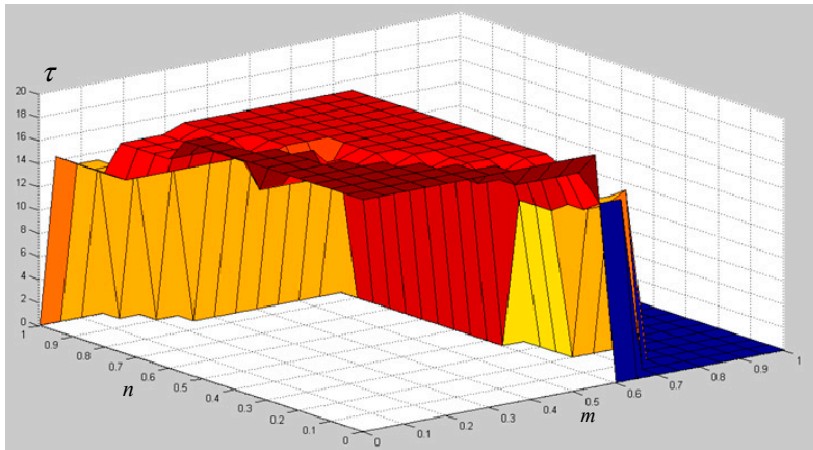

**Figure 10.** The relationship between maximum allowable time-delay $\tau$ to make the system stable and control coefficients $m$, $n$ (sampling time equals to 20 ms).

*4.2. Stability Analysis of Data Dropout*

Data-dropout is very likely to occur during the network transmission and then influence the system stability and performance especially in high-speed transmission having channel congestion. The state function containing data-dropout can be written by

$$\hat{x}(k) = \theta_k x(k) + (1 - \theta_k)\hat{x}(k-1), \tag{18}$$

where $\theta_k \in \{0,1\}$ represents the occurrence of data dropout. In this sense, $\theta_k = 0$ means data-dropout entirely occurs and $\theta_k = 1$ indicates there is no data-dropout. $\hat{x}(k)$ is the state signal received by controller at time of $kh$. The control feedback is $u(k) = K\hat{x}(k)$, the state Equation (18) is rewritten by

$$\hat{u}(k) = \theta_k K x(k) + (1 - \theta_k)K\hat{x}(k-1). \tag{19}$$

Combining Equations (A19), (14), and (19), the state-space solution can be derived as

$$x(k+1) = [e^{Ah} + \theta_k \Gamma_0 BK]x(k) + (1 - \theta_k)\Gamma_0 BK\hat{x}(k-1), \tag{20}$$

where $\Gamma_0 = \Gamma_1(0) = \int_0^h e^{As}ds$, $h$ is the sampling period.

Defining $z(k) = \left[x^T(k), \hat{x}^T(k-1)\right]^T$, Equation (20) can be transformed into

$$z(k+1) = G(\theta_k)z(k), \tag{21}$$

where $G(\theta_k) = \begin{bmatrix} e^{Ah} + \theta_k \Gamma_0 BK & (1 - \theta_k)\Gamma_0 BK \\ \theta_k I & (1 - \theta_k)I \end{bmatrix}$.

Because $\{\theta_k, k = 1, 2 \cdots\}$ is an ergodic sequence and has only two values, 0 and 1, we define $\theta_0$ as occurrence probability of data dropout

$$\theta_0 = \lim_{T \to \infty} \frac{1}{T}\sum_{k=0}^{T} \theta_k.$$

Obviously $0 < \theta_0 < 1$, Equation (20) is expressed as

$$z(k+1) = \theta_k G_0 z(k) + (1 - \theta_k)G_1 z(k), \tag{22}$$

where $G_0 = \begin{bmatrix} e^{Ah} + \Gamma_0 BK & 0 \\ I & 0 \end{bmatrix}$, $G_1 = \begin{bmatrix} e^{Ah} & \Gamma_0 BK \\ 0 & I \end{bmatrix}$.

One data dropout stability theorem of the network-based control system with certainly discrete-time model and single-packet transmission was proposed in [30], which gave the sufficient condition by means of linear control theory. Omitting the derivation, the analysis result and calculation steps are described as follows:

(1) If the hypothesis of a series of nonlinear equation construction is true, and there is

$$\lambda_1^{\theta_0} \lambda_2^{(1-\theta_0)} < 1,$$

then the system described by Equation (22) is asymptotically stable and almost each solution is converging to the origin.

(2) If $(A, B)$ and $(e^{Ah}, \Gamma_0 B)$ is controllable, linear control theory can be used to design a matrix $K$ to make the characteristic roots of both $e^{Ah} + \Gamma_0 BK$ and $G_0$ inside the unit circle. Consequently, there exists a positive matrix $P$ to satisfy the condition of $G_0^T P G_0 - \lambda_1 P < 0$ under $\lambda_1 \in [0, 1)$, where

$$\lambda_2 = \lambda_{\max}(P^{-\frac{1}{2}} G_1^T P G_1 P^{-\frac{1}{2}}). \tag{23}$$

In order to gain a better understanding, $\theta_d$ is employed to represent the occurrence rate of data dropout. In this sense, $\theta_d = 1$ indicates there is no data dropout, whereas all transmitted data is missing or in incorrect reception when $\theta_d = 0$. According to the above stability criterion, the bounds of maximum occurrence probabilities of data drop-out can be calculated, the results are shown in Figure 11. This is a two-dimension curve, plotted by $\lambda_1$ on the horizontal axis and $\theta_d$ on the vertical. The waveform shows a nonlinear downward trend with the increasing of $\lambda_1$. If $\lambda_1 \to 0$, the maximum occurrence rate of data dropout $\theta_d$ is approaching to 0.45. When $\lambda_1 \to 1$, the value of $\theta_d$ gets very close to zero. In general, more accurate predictions can be achieved by seeking results of $\lambda_1$ in the interval of (0.3, 0.7). Take the median of $\lambda_1 = 0.5$, the bound of maximum occurrence rate $\theta_d$ can be obtained by 0.118.

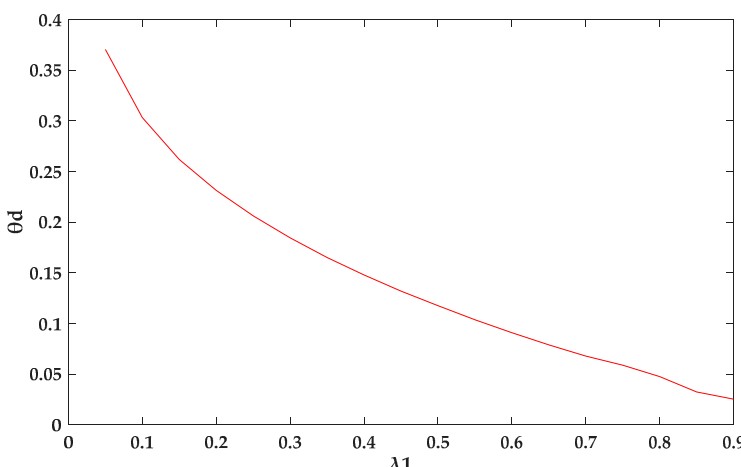

**Figure 11.** Relationship diagram of occurrence rate bounds of data dropout $\theta_d$ and variant $\lambda_1$.

*4.3. Stability Analysis of System with Small Time-Scale*

One of the prerequisites about the outcomes obtained by above mathematical analysis is that the sampling period is relatively large (20 ms), which means the burden the control brings to the network is small and we can use a low-bandwidth network to finish the job. It is necessary to test the performance under the circumstances with fast-transmission of the network to adapt to different system time-scales.

The situation of maximum time-delay less than *h* is still considered in this case. Considering *h* belongs to microseconds level, the bandwidth of the network will be bigger than the case with *h* = 20 ms. The relationship among control coefficients *m*/*n* and time-delay maximum bound $\tau$ values is presented in Figure 12. Through this three-dimension diagram, the selection scope of allowable time-delay to make system stable in most occasions is quite large, which shows a good robustness of time.

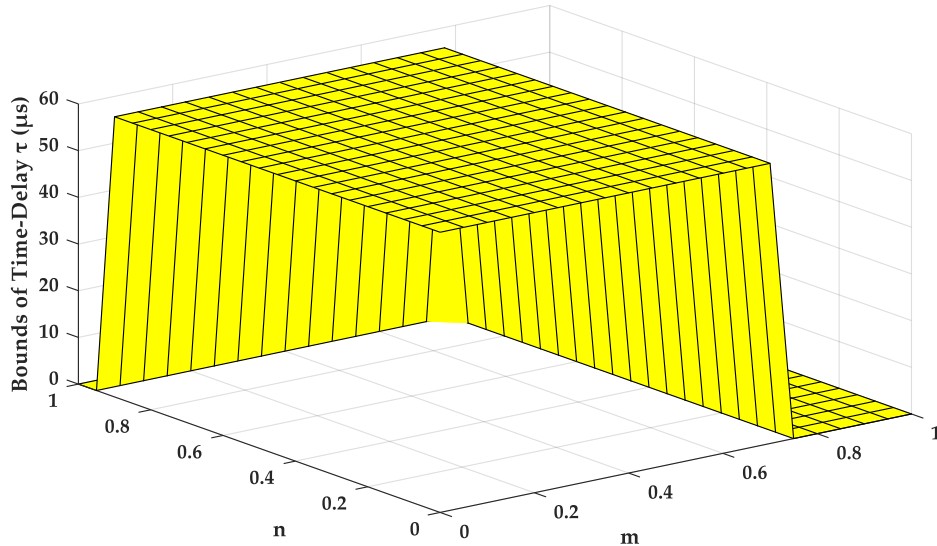

**Figure 12.** The relationship between maximum allowable time-delay $\tau$ and control coefficients *m*, *n* (sampling time *h* equals 60 μs).

## 5. Experimental Results

In order to verify the accuracy of mathematical analysis and the effectiveness of power allocation strategy, an experimental platform consisting of two 3 kW inverters using three-phase three-wire half-bridge topology was built, the architecture diagram is shown in Figure 13. Each inverter was connected via CAN bus, which has perfect priority arbitration and enough transmission speed to form an information-shared network with other inverters. DSP (Digital Signal Processor) TMS320F28335 was applied to realize full digital control. The inductive current and capacitor voltage were converted into digital values after processing by the ADC (Analog-to-Digital) module embedded in the DSP, and abc-dq0 transformation was carried out. Through the instantaneous power calculation and low-pass filtering, the active power and reactive power $P_{CALn}$ and $Q_{CALn}$ (*n* = 1,2 in this experimental platform) were obtained, then they were put into the weighting algorithm with the network power information $P_{COMn}$ and $Q_{COMn}$ to realize the P/Q droop control strategy. The transmission was set as bi-directional mode, the synchronous ID was equipped for each transmitted data to ensure all the communication action and control process have the same timestamp. The sampling period of this network-based control strategy was originally set as 20 ms. The specification parameters of this platform are shown in Table 2.

The control of this network-based system can be analyzed through three loops. First of all, after the instantaneous active and reactive power calculation based on dq0 coordinate transformation theory being done, P/Q droop control is designed to produce the voltage reference and provide the regulation of the output frequency. The main function of P/Q droop control is to shape a desired steady-state and dynamic performance, guarantee the system in an ideal power-sharing feature. Secondly, the output voltage is closely tracked through the voltage loop to ensure the stability of the inverter. Finally, the current loop is used to reflect the state change of the system and improve the dynamic response speed. A physical picture of the experimental platform is captured in Figure 14, where

two three-phase inverters are operated in parallel, the network interconnection, the white line in Figure 14a, is used to realize the network-based control. IPM (Intelligent Power Module) PM75RLA060 is employed as a main-circuit module of inverters. The output of each inverter is connected to three-phase transformers as shown in Figure 14b, which is designed as Y/△-1 connection for primary and secondary winding. Inductance L/capacitor C filter is employed, and the filter inductance L is obtained by leakage inductance of the three-phase transformer. The outputs of the three-phase transformer are linked with the common three-phase load via contactors as shown in Figure 14b.

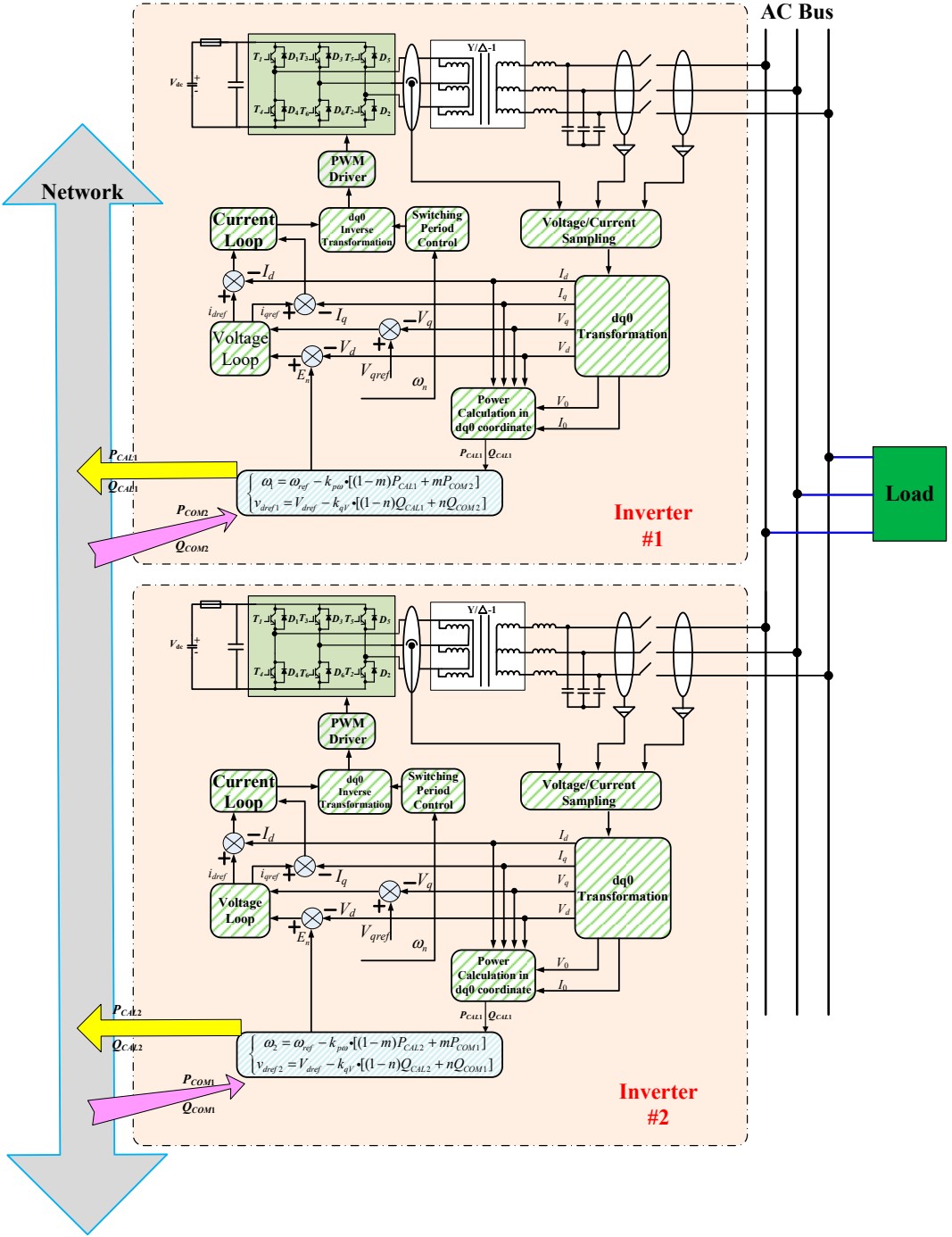

**Figure 13.** Experimental architecture diagram of the network-based control system for parallel operation with two three-phase inverters.

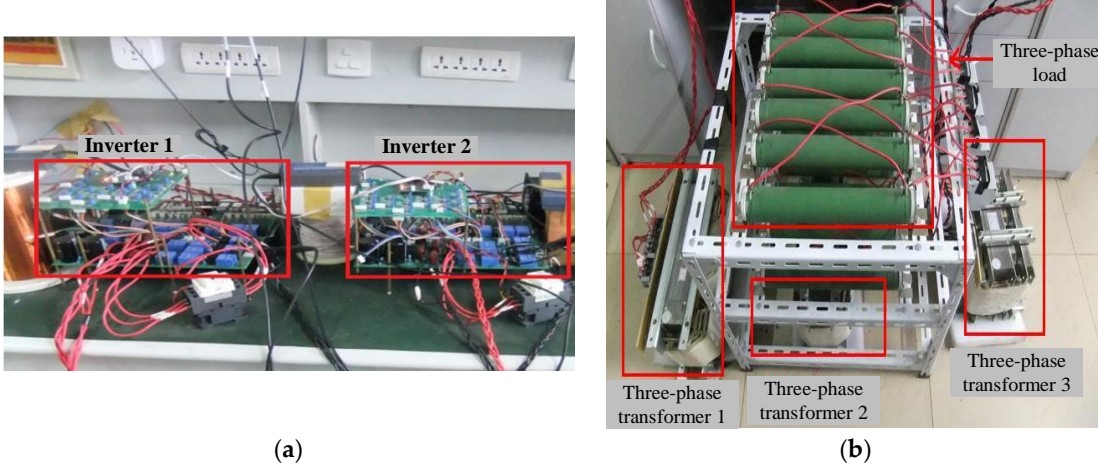

**Figure 14.** The experimental environment for the network-based control system with two paralleled three-phase inverters. (**a**) Two paralleled inverters. (**b**) Three-phase load and transformers.

The steady-state output voltage and current in three different resistance loads when two inverters being paralleled are shown in Figure 15a–c. It is clearly shown that when the load changes, the A-B-phase output line voltage of the two inverters ($v_{AB1}$, $v_{AB2}$) basically remains unchanged, while the inductive currents (i.e., A-phase output current $i_{oa1}$, $i_{oa2}$) have the constant proportional variation. In addition, the difference value of these two output currents, i.e., circulating current is nearly zero, which means the load-sharing is quite satisfactory as expected.

To better describe the superiority of the network-based strategy of inverter parallel operation in steady-state accuracy. The load-sharing error is defined and introduced by

$$\left| (I_{oa1} - (I_{oa1} + I_{oa2})/2)/((I_{oa1} + I_{oa2})/2) \right| \times 100\%. \tag{24}$$

Output current RMS (Root Mean Square) of each inverter was measured and recorded simultaneously. To make the results more reliable and precise, a number of groups of data were collected and put into the load-sharing error equation (Equation (24)). Then these errors are compiled as statistical data, such as average error or maximum/minimum error, etc. The inverter-parallel experiments were carried out by using network-based control and traditional power droop control (Equation (6)). The load-sharing errors can also be statistically processed and compared to quantify the performance. The average load-sharing errors by using the network-based control and traditional power droop control are addressed in Figure 16. It is clearly indicated that on most occasions, the average load-sharing errors achieved by using network-based control are less than those with the conventional droop control method. More importantly, these errors can be restricted within 5% even though the network-based method must tolerate inherent network-induced time-delay (the delay is much less than one sampling period 20 ms without considering an unexpected and abnormal communication situation).

*5.1. Impact of Time-Delay*

To investigate the time-delay impact, we artificially produced different transmission times and observed how much time-delay would influence the system performance. As mentioned before, the timestamp will be labeled by ID for each transmitted power data for each inverter, which means synchronous function for the digital control is required. When the start-time begins with the power data of the other inverter arriving at the CAN bus toolbox, the timing of control time-delay can be calculated. In general, if the sampling period is long enough so that the controller can await the transmitted data with the same ID, the network-induced time-delay has little effect on the system stability.

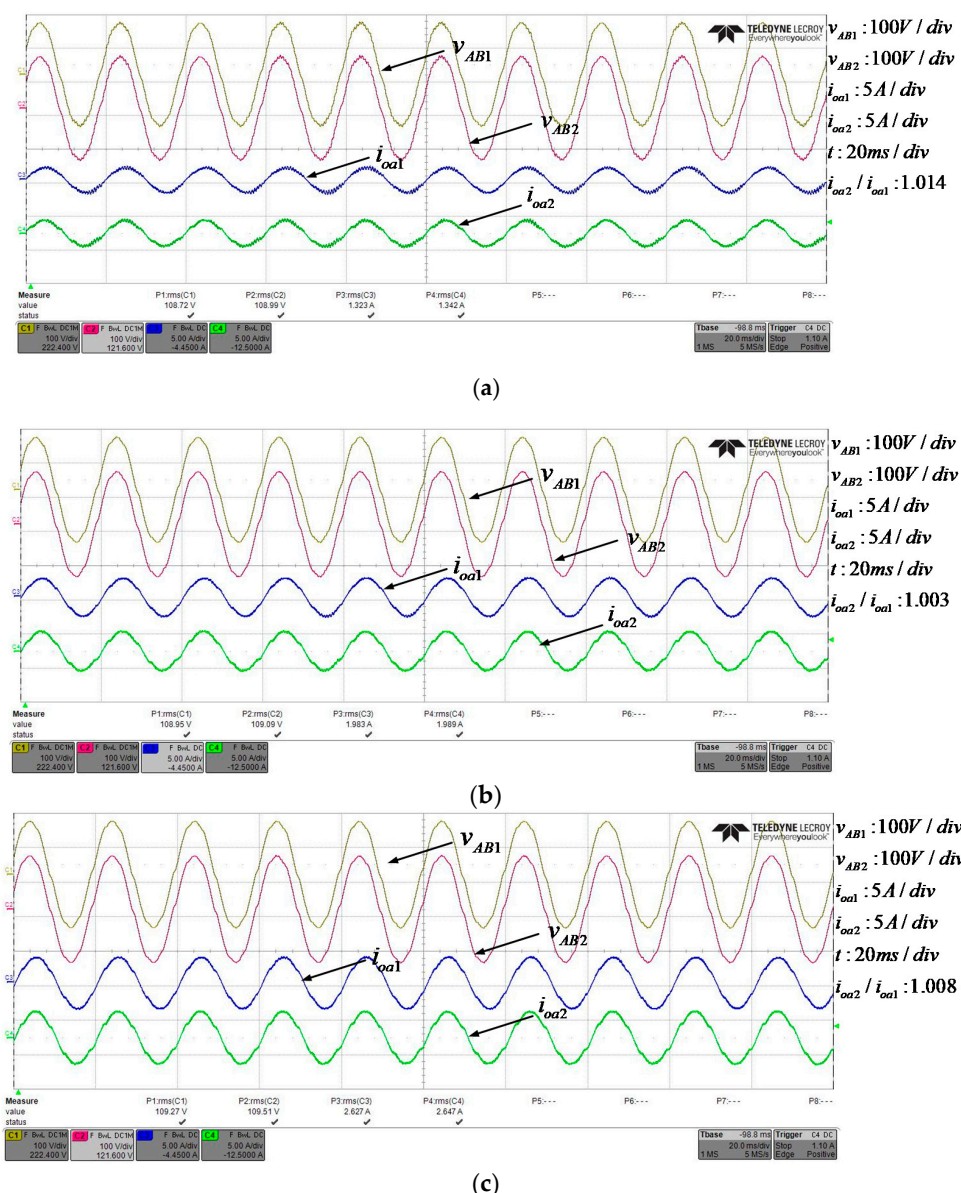

**Figure 15.** Steady-state output of inverter parallel system (A-phase) based on network-based control in three different resistance loads. (**a**) $R_L = 70 \; \Omega$; (**b**) $R_L = 47 \; \Omega$; (**c**) $R_L = 35 \; \Omega$.

One group control coefficients $m = n = 0.6$ was used to test the time-delay impact. In the theoretical analysis in Section 4.1, the allowable time-delay bound can be reached at one sampling period $h = 20$ ms in a wide $m/n$ range including $m = n = 0.6$. The time-delay artificially made by the time buffer was created at $\tau = 20$ ms. The experimental results are shown in Figure 17, where output current $i_{oa1}$, $i_{oa2}$ (the upper two) and voltage waveforms $v_1$ (the lower green one) in load resistance 35 $\Omega$, 47 $\Omega$, and their mutual dynamic transit are displayed. These results show a desired steady-state output and very small circulating current between these inverters, an excellent dynamic response for load step changes in a network-based control strategy. To further describe its delay-insensitive feature, 30 load-sharing errors calculated by 30 groups of output current RMS measured at 5 min intervals during steady-state operation were recorded and numerically processed by average value. In this case, the average load-sharing error is within 2.8% when the load resistance is 35 $\Omega$, which indicates that the excellent system performance is still maintained even when the time-delay is artificially increased to 20 ms. It is an inspiring outcome because the network-based control demonstrates a good robustness towards network-induced time-delay.

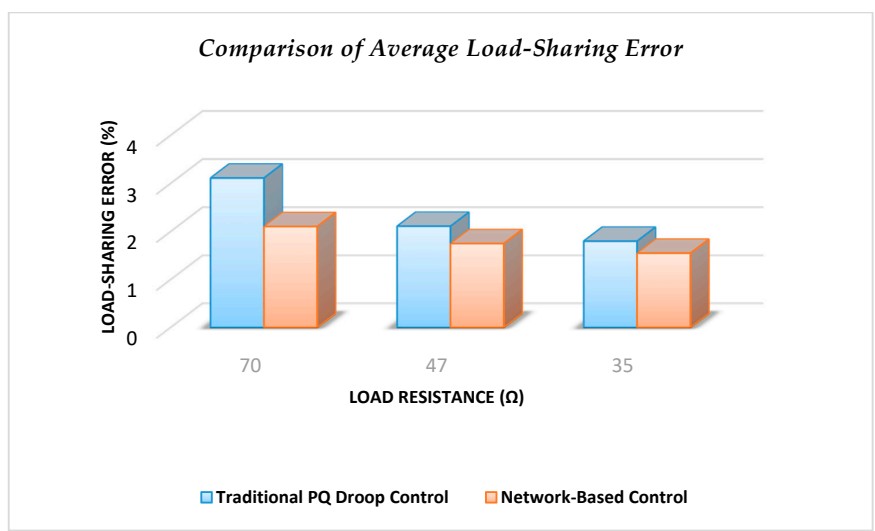

**Figure 16.** The average load-sharing errors of three different load resistances using traditional power droop control and network-based control.

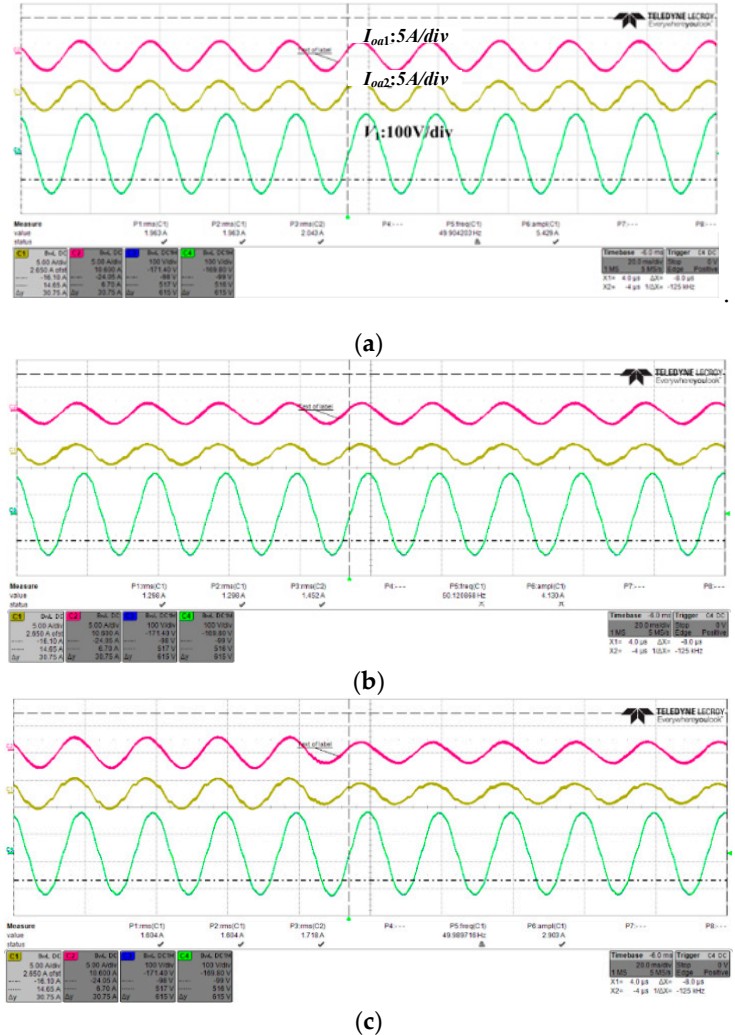

**Figure 17.** Output current and voltage of two paralleled inverter system (A-phase) based on network-based control. (**a**) Load resistance is 35 Ω, (**b**) load resistance is 47 Ω, (**c**) load resistance is switched from 35 Ω to 47 Ω.

### 5.2. Impact of Data Dropout

In this experimental test, $\theta_d$ is defined to quantify the data dropout. For convenience, it can be designed as constant packet loss ratio. For example, $\theta_d = 0.4$ represents that 6 out of 10 data have been discarded or unsuccessfully transmitted. In actual operation, the deliberate design of $\theta_d$ is based on the transmitted ID of power data. For each 10 continuous data transmissions with contiguous IDs, we can artificially choose some of them out of the network-based control to fit the $\theta_d$ value. The comparisons of output current with inverter parallel operation between $\theta_d = 1$ and $\theta_d = 0.4$ in two load conditions can be observed in Figure 18. As described above, $\theta_d = 1$ means there is no data dropout during the data transmission. From the results we can see power-sharing performance is barely affected by data dropout even if almost half of the transmission data is missing or blocked from being received. In addition, we tested worse network conditions such as $\theta_d = 0.3$ and $\theta_d = 0.25$, and the system kept stable and had satisfactory load-sharing performances as well. However, there will be a significant increase of load-sharing error when more data dropouts occur, especially when $\theta_d < 0.2$, which is largely consistent with the theoretical analysis in Section 4.2. Overall, the network-based control shows a strong robustness towards data dropout.

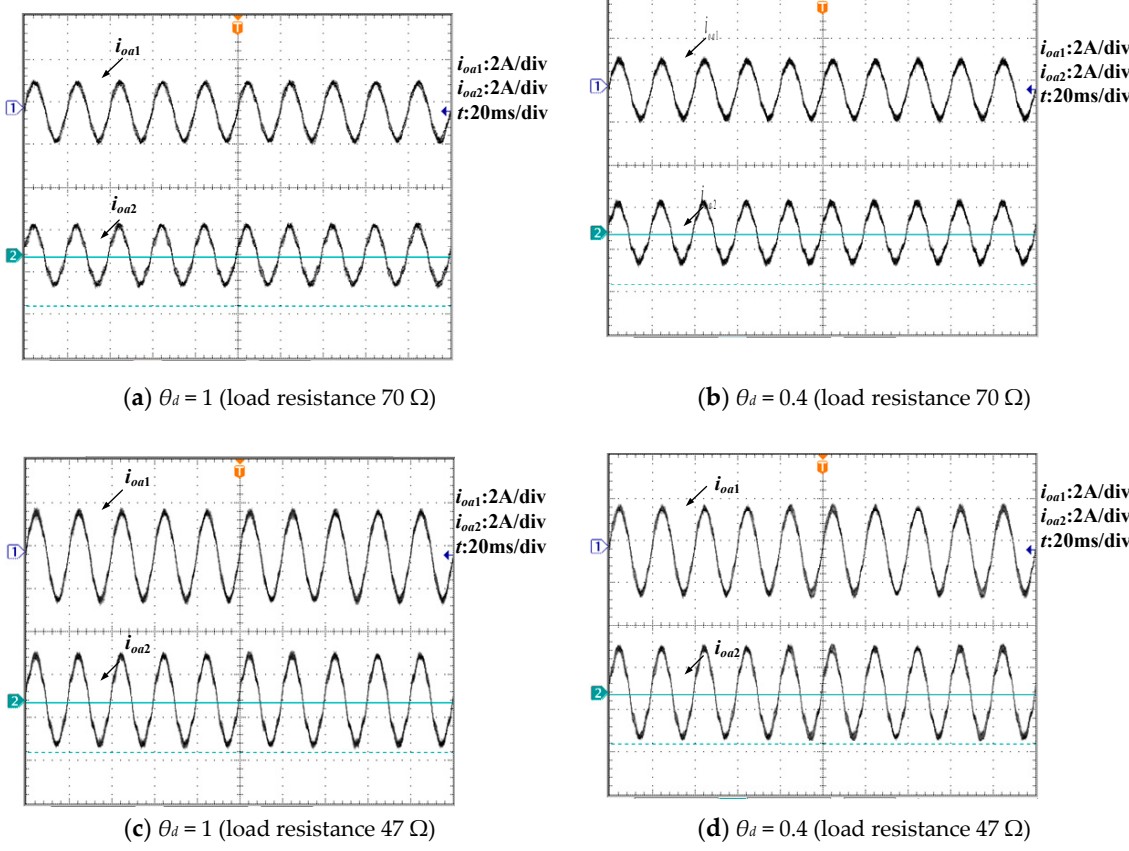

(**a**) $\theta_d = 1$ (load resistance 70 $\Omega$)  (**b**) $\theta_d = 0.4$ (load resistance 70 $\Omega$)

(**c**) $\theta_d = 1$ (load resistance 47 $\Omega$)  (**d**) $\theta_d = 0.4$ (load resistance 47 $\Omega$)

**Figure 18.** Output current of paralleled inverter by using a network-based control method when $\theta_d = 1$ and $\theta_d = 0.4$ respectively.

### 5.3. Performance Tests of Small Time-Scale

The adaptation of network-based control with a small time-scale is investigated in this section. We designed the sampling time $h$ approaching to microsecond level, 60 µs and the time-delay of transmission was set as $h$. The steady-state and dynamic performance with load resistance 70 $\Omega$ are shown in Figure 19. From the output current of inverter #1 and #2, $i_{o1}$ and $i_{o2}$, it demonstrates that although there exists some distortion and oscillation in the steady-state and dynamic state operation, the overall results are satisfactory. The existing output-current distortion is potentially due to the fact

that the voltage reference is frequently updated upon the arrival of fast-speed transmission data, and the error of sampling or calculation would increase the possibility of impacting the voltage, current control loop, and power quality. To give a further evaluation, the average load-sharing errors are collected and calculated. Even when the time-delay is up to 100 μs, the load-sharing is still kept within 5%. In brief, the long-term good reliability, excellent control accuracy, strong robustness and wide time-scale compatibility can be guaranteed by means of a network-based control strategy.

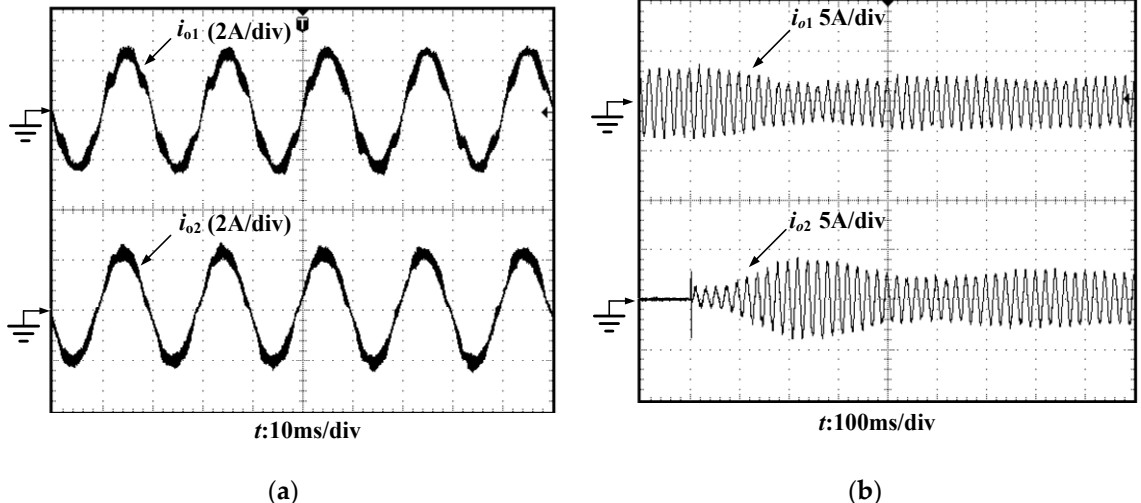

**Figure 19.** Output current of paralleled inverters when the sampling period *h* is 60 μs and the time-delay is artificially set as *h*. (**a**) Steady-state of output current $i_{o1}$ and $i_{o2}$; (**b**) dynamic-state when inverter #2 is thrown into parallel system.

## 6. Extended Discussion of Network-Based Control Method

To further improve the load-sharing performance of inverter parallel operation in AC microgrids, the presented network-based control can be extended to application in different kinds of scenarios, including the plug-and-play pathway, the ability to diagnose and deal with the transmission failure.

### 6.1. Considering the Rated Power

In order to enhance the expandability of network-based control, the rated power of each inverter should be considered, which means the load-sharing can be proportionally implemented based on their rated power. The network-based control strategy can be modified as follows based on control Equations (11)–(13):

$$\begin{cases} \omega_1 = \omega_{ref} - k_{p\omega} \cdot [(1 - m_1 - m_2)P_{CAL1} + m_1 P_{COM2}/\eta_{12} + m_2 P_{COM3}/\eta_{13}] \\ v_{dref1} = V_{dref} - k_{qV} \cdot [(1 - n_1 - n_2)Q_{CAL1} + n_1 Q_{COM2}/\lambda_{12} + n_2 Q_{COM3}/\lambda_{13}] \end{cases}, \tag{25}$$

$$\begin{cases} \omega_2 = \omega_{ref} - (k_{p\omega}/\eta_{12})[(1 - m_1 - m_2)P_{CAL2} + m_1 P_{COM1}\eta_{12} + m_2 P_{COM3}\eta_{23}] \\ v_{dref2} = V_{dref} - (k_{qV}/\lambda_{12})[(1 - n_1 - n_2)Q_{CAL2} + n_1 Q_{COM1}\lambda_{12} + n_2 Q_{COM3}/\lambda_{23}] \end{cases}, \tag{26}$$

$$\begin{cases} \omega_3 = \omega_{ref} - (k_{p\omega}/\eta_{13})[(1 - m_1 - m_2)P_{CAL3} + m_1 P_{COM1}\eta_{13} + m_2 P_{COM2}\eta_{23}] \\ v_{dref3} = V_{dref} - (k_{qV}/\lambda_{13})[(1 - n_1 - n_2)Q_{CAL3} + n_1 Q_{COM1}\lambda_{13} + n_2 Q_{COM2}\lambda_{23}] \end{cases}, \tag{27}$$

where $\eta_{12} = P_2/P_1 = \eta_2/\eta_1$, $\lambda_{12} = Q_2/Q_1 = \lambda_2/\lambda_1$, $\eta_{13} = P_3/P_1 = \eta_3/\eta_1$, $\lambda_{13} = Q_3/Q_1 = \lambda_3/\lambda_1$, $\eta_{23} = P_3/P_1 = \eta_3/\eta_1$, $\lambda_{23} = Q_3/Q_2 = \lambda_3/\lambda_2$. $\eta_2$ and $\lambda_i$ are the power weighting coefficients of inverter *i*, relative coefficients $\eta_{ij}$ and $\lambda_{ij}$ are obtained by the ratio of rated power of inverter *j* and *i*. Then the data label ID, transmission priority ID, active and reactive power, and the power weighting coefficients are sent to other inverters via network. The power ratings among the inverters can be

designed to be flexible and change quickly based on the requirements. The dynamic performance with the variant loads can be in a rapid response by means of the network.

To verify the validity of the proposed method, the test simulation was implemented with three three-phase paralleled inverters by means of MATLAB and is shown in Figure 20, where $V_{dref}$ = 311V, the ratio of power-sharing based on the different rated power of three inverters is $\eta_1 : \eta_2 : \eta_3 = 1 : 2 : 3$, and the three inverters are put into the parallel operation supplying resistance load at the time $t$ = 1 s simultaneously. The time-delay is artificially set as 10 ms between adjacent transmission. The results of output current ($i_{o1}$, $i_{o2}$, and $i_{o3}$) and output active power ($P_1$, $P_2$, and $P_3$) show that an excellent dynamic response of the network-based control method at switching load condition, and the load-sharing is strictly corresponding to the load-sharing ratio (1:2:3) as expected. In a sense, the network-based control can be designed flexibly according to the actual application requirements.

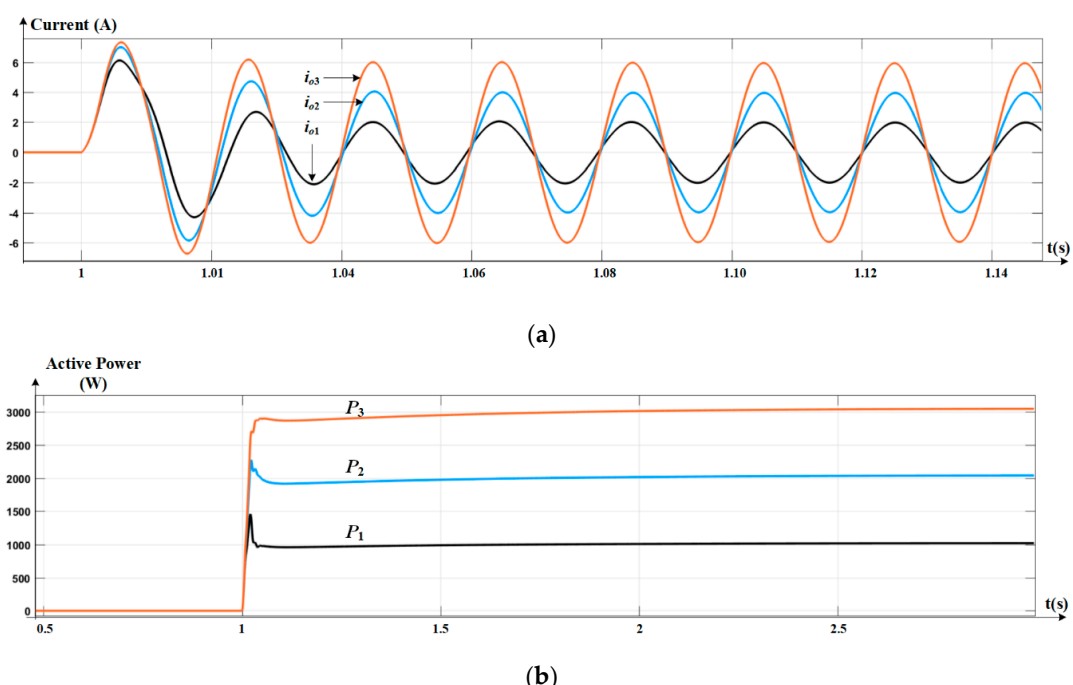

(a)

(b)

**Figure 20.** Output current and output active power of paralleled inverters when load is proportionally allocated based on their rated power (the ratio of $\eta_1 : \eta_2 : \eta_3 = 1 : 2 : 3$). (**a**) Output current $i_{o1}$, $i_{o2}$, and $i_{o3}$; (**b**) output active power $P_1$, $P_2$, and $P_3$.

## 6.2. Considering the Difference of Time-Delay Level

As analyzed, the network-based control has a wide time-scale compatibility, which ranges from hundreds of microseconds to milliseconds. In addition, the time-scale levels are differently distributed in three control loops, current, voltage, and power control loop, as shown in Figure 21. The impact of the communication time-delay on system stability is mainly dependent on the power control loop since the network data is acquired and exchanged with the power control loop, which has a quite slow time-scale level. The implementation of network-based control is flexible. For example, even the time-delay levels of the inverters are different, their network-based time-scales could be unified on the same level. If the time-delay of each inverter could not be changed, the network-based strategy would be still available by the MAC (Medium Allocation Control) of the network to make sure the inverters are synchronized in their independent steps. As for what plan should be pursued, the basic rule is to guarantee the system stability and enhancement of the parallel operation. Since the differences of rated power and time-delay level among inverters can be thoughtfully considered, the plug-and-play function of the inverter parallel system with network-based control will be more achievable.

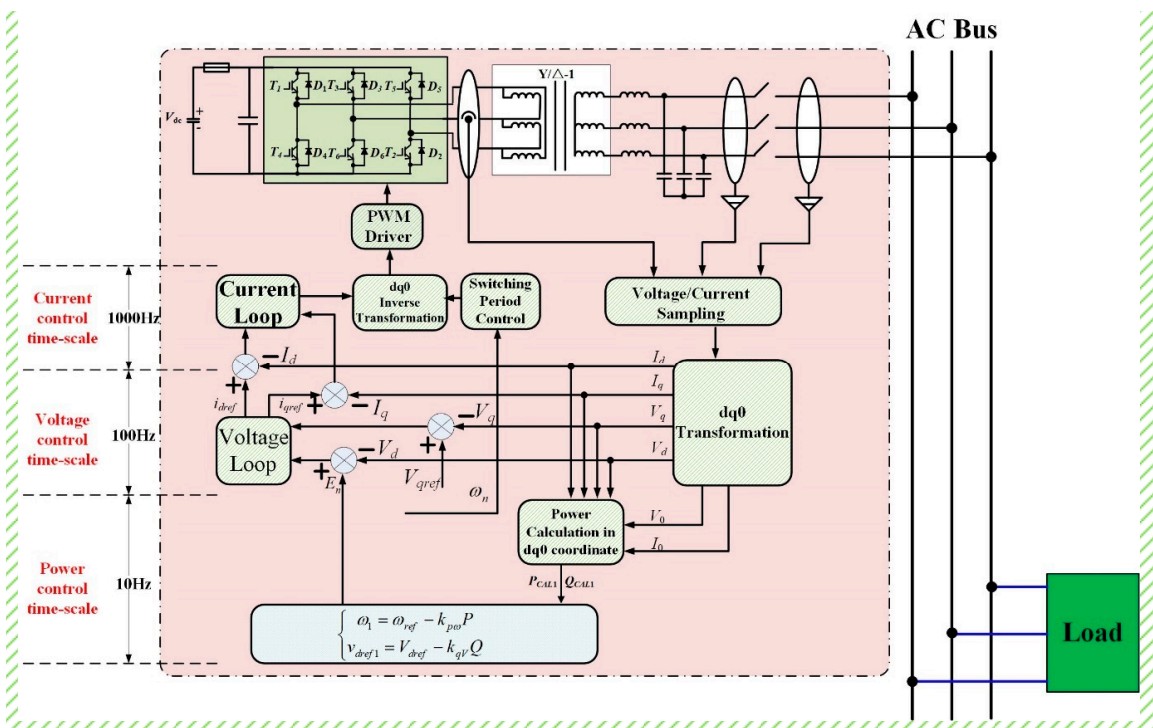

**Figure 21.** The different control time-scale distribution of current, voltage, and power control loop in a typical droop-controlled AC microgrid.

*6.3. Considering the Abnormal Situation of Communication*

Another important issue, the abnormal situation or failure of communication should be considered. In this case, the adaptive data-processing method is proposed to prevent this kind of failure, as shown in Figure 22. Take one inverter as an analysis example, the "Network Data Analyzer" is used to detect the communication condition during the transmission. The "State-Determining Logic of Network Condition" part is used to judge whether or not the network is normal. If the answer is "Yes", which means the communication is running in a good way, the network-based control continues. To the contrary, if "No" is obtained, the network-based control is aborted, and the control flow is switched into traditional power droop control. The green lines are addressed as network data and red one is its own power data via on-line calculation. Whatever the control direction the inverter follows, the derived control results have to be the input of voltage and current control. It is necessary to mention, the "Network Data Analyzer" is kept working continuously to diagnose the communication fault precisely. The event-triggered function is turned on when the results are finally definite.

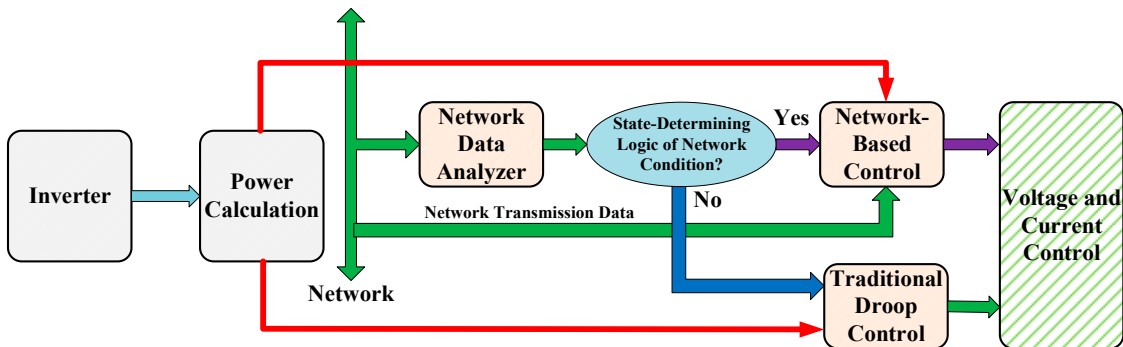

**Figure 22.** The adaptive data-processing method of network-based control to deal with the abnormal communication situation.

## 7. Conclusions

The AC microgrids (MGs) are capable of operating in grid-connected and islanded mode, handling their transitions. For islanded mode, droop-controlled inverters and their parallel operation are employed to balance the generated active and reactive power in the microgrid with the demand of local loads to ensure system stability. Different from traditional power droop control, network-based control was presented in this paper to obtain the potential performance of paralleled inverters. However, the negative factors derived from network such as time-delay and data dropout would have unexpected impacts on system, may degrade or influence the system stability and performance. From this perspective, the superiority of network-based control was analyzed. System models with time-delay and data dropout were built to evaluate their impacts. In addition, the same procedure was carried out to test the outcomes when different time-delay and data dropout are introduced. It was observed from the results that the network-based control has satisfactory performance under these unusual scenarios and strong robustness to these negative network-induced factors. On the other hand, if we change the control period and time-scale by increasing the sampling period from hundreds of microseconds to milliseconds level, system stability is still maintained in a very wide range with varied control coefficients, which means the system has a wide time-scale compatibility. In this sense, it provides the possibility of communication integration for different control layers in AC microgrid design with hierarchical structure. Experimental results verified the effectiveness of the analytical method, and the strong robustness and good time-scale compatibility of the network-based control strategy in a droop-controlled AC microgrid system. Finally, some important issues such as variant rated power and time-delay level existing in the paralleled inverters and communication failure were addressed, the description of problem-solving and explanation were also provided.

**Author Contributions:** Conceptualization, Y.Z. and F.Z.; methodology, Y.Z.; software, Y.Z.; validation, Y.Z., F.Z. and Y.Q.; formal analysis, Y.Z.; investigation, Y.Z.; resources, Y.Z., F.Z. and P.Z.; data curation, Y.Z.; writing—original draft preparation, Y.Z.; writing—review and editing, Y.Z.; project administration, Y.Z.; funding acquisition, Y.Z. All authors have read and agreed to the published version of the manuscript.

**Funding:** This research was funded by the National Natural Science Foundation of China under grant 51877058, in part by Natural Science Foundation of Zhejiang Province under grant LY18E070004, LQ18E070001 and Scientific and Technical Projects of State Grid Zhejiang Electric Power Company, LTD. under grant 5211HZ1800UM.

**Conflicts of Interest:** The authors declare no conflict of interest.

## Appendix A

*Modeling of The Network-Based Control Method*

The model of network-based control method can be likewise deduced and fully presented for the analysis of time-delay and data dropout.

To facilitate analysis, Figure A1 is a simplified circuit of three-phase inverter parallel system. As shown, $V_{on}$ is the actual output voltage amplitude of AB-phase for inverter $n$. If there is no time-delay and other network issues, then we have $P_{CALn} = P_{COMn} = p_n$, $Q_{CALn} = Q_{COMn} = q_n$ in Equations (16) and (17). In addition, the control variables are defined as follows,

$$\begin{cases} p_{N1} = (1-m)p_1 + mp_2 \\ p_{N2} = (1-m)p_2 + mp_1 \\ q_{N1} = (1-n)q_1 + nq_2 \\ q_{N2} = (1-n)q_2 + nq_1 \end{cases}. \tag{A1}$$

The network-based control method (16) and (17) can be obtained by

$$\begin{cases} \omega_1 = \omega_{ref} - k_{p\omega}p_{N1} \\ v_{dref1} = V_{dref} - k_{qV}q_{N1} \\ \omega_2 = \omega_{ref} - k_{p\omega}p_{N2} \\ v_{dref2} = V_{dref} - k_{qV}q_{N2} \end{cases}.$$ (A2)

Taking the parallel system of two inverters as an example and combining Equations (7), (8), and (A2), the control equation can be written as

$$\begin{cases} \frac{d\omega_1}{dt} = -\omega_f\omega_1 + \omega_f\omega_{ref} - k_{p\omega}\omega_f p_{N1} \\ \frac{dV_{o1}}{dt} = -\omega_f v_{o1} + \omega_f V_{ref} - k_{qV}\omega_f q_{N1} \\ \frac{d\omega_2}{dt} = -\omega_f\omega_2 + \omega_f\omega_{ref} - k_{p\omega}\omega_f p_{N2} \\ \frac{dV_{o2}}{dt} = -\omega_f v_{o2} + \omega_f V_{ref} - k_{qV}\omega_f q_{N2} \end{cases}.$$ (A3)

In dq0 coordinate, the output voltage vector of inverter can be represented as

$$\vec{V}_{on} = v_{dn} + jv_{qn},$$

where

$$V_{on} = \left|\vec{V}_{on}\right| = \sqrt{v_{dn}^2 + v_{qn}^2} = \frac{v_{dn}^2 + v_{qn}^2}{\sqrt{v_{dn}^2 + v_{qn}^2}} = \frac{v_{dn}}{\sqrt{v_{dn}^2 + v_{qn}^2}} \cdot v_{dn} + \frac{v_{qn}}{\sqrt{v_{dn}^2 + v_{qn}^2}} \cdot v_{qn} = m_{dn}v_{dn} + m_{qn}v_{qn},$$ (A4)

$$\begin{cases} v_{dn} = V_{on}\cos\varphi_n \\ v_{qn} = V_{on}\sin\varphi_n \\ \varphi_n = \arctan(v_{qn}/v_{dn}) \end{cases},$$ (A5)

and

$$m_{dn} = \frac{v_{dn}}{\sqrt{v_{dn}^2 + v_{qn}^2}}, \ m_{qn} = \frac{v_{qn}}{\sqrt{v_{dn}^2 + v_{qn}^2}},$$

so

$$\frac{dV_{on}}{dt} = \frac{2v_{dn}}{2\sqrt{v_{dn}^2 + v_{qn}^2}} \cdot \frac{dv_{dn}}{dt} + \frac{2v_{qn}}{2\sqrt{v_{dn}^2 + v_{qn}^2}} \cdot \frac{dv_{qn}}{dt} = m_{dn}\frac{dv_{dn}}{dt} + m_{qn}\frac{dv_{qn}}{dt}.$$ (A6)

In addition,

$$\omega_n = \frac{d\varphi_n}{dt} = \frac{1}{1 + (\frac{v_{qn}}{v_{dn}})^2} \cdot (-\frac{v_{qn}}{v_{dn}^2} \cdot \frac{dv_{dn}}{dt} + \frac{1}{v_{dn}} \cdot \frac{dv_{dq}}{dt}) = -\frac{v_{qn}}{v_{dn}^2 + v_{qn}^2} \cdot \frac{dv_{dn}}{dt} + \frac{v_{dn}}{v_{dn}^2 + v_{qn}^2} \cdot \frac{dv_{qn}}{dt} = f_{dn}\frac{dv_{dn}}{dt} + f_{qn}\frac{dv_{qn}}{dt},$$ (A7)

where

$$f_{dn} = -\frac{v_{qn}}{v_{dn}^2 + v_{qn}^2}, \ f_{qn} = \frac{v_{dn}}{v_{dn}^2 + v_{qn}^2}.$$

In combination with Equations (A1)–(A7), the solution can be obtained in the form of matrix

$$
\begin{bmatrix}
\frac{d\omega_1}{dt} \\
\frac{dv_{d1}}{dt} \\
\frac{dv_{q1}}{dt} \\
\frac{d\omega_2}{dt} \\
\frac{dv_{d2}}{dt} \\
\frac{dv_{q2}}{dt}
\end{bmatrix}
=
\begin{bmatrix}
-\omega_f & 0 & 0 & 0 & 0 & 0 \\
\frac{m_{q1}}{f_{d1}m_{q1}-f_{q1}m_{d1}} & \frac{f_{q1}m_{d1}\omega_f}{f_{d1}m_{q1}-f_{q1}m_{d1}} & \frac{f_{q1}m_{q1}\omega_f}{f_{d1}m_{q1}-f_{q1}m_{d1}} & 0 & 0 & 0 \\
\frac{m_{d1}}{f_{q1}m_{d1}-f_{d1}m_{q1}} & \frac{f_{d1}m_{d1}\omega_f}{f_{q1}m_{d1}-f_{d1}m_{q1}} & \frac{f_{d1}m_{q1}\omega_f}{f_{q1}m_{d1}-f_{d1}m_{q1}} & 0 & 0 & 0 \\
0 & 0 & 0 & -\omega_f & 0 & 0 \\
0 & 0 & 0 & \frac{m_{q2}}{f_{d2}m_{q2}-f_{q2}m_{d2}} & \frac{f_{q2}m_{d2}\omega_f}{f_{d2}m_{q2}-f_{q2}m_{d2}} & \frac{f_{q2}m_{q2}\omega_f}{f_{d2}m_{q2}-f_{q2}m_{d2}} \\
0 & 0 & 0 & \frac{m_{d2}}{f_{q2}m_{d2}-f_{d2}m_{q2}} & \frac{f_{d2}m_{d2}\omega_f}{f_{q2}m_{d2}-f_{d2}m_{q2}} & \frac{f_{d2}m_{q2}\omega_f}{f_{q2}m_{d2}-f_{d2}m_{q2}}
\end{bmatrix}
\begin{bmatrix}
\omega_1 \\
v_{d1} \\
v_{q1} \\
\omega_2 \\
v_{d2} \\
v_{q2}
\end{bmatrix}
$$
$$
+
\begin{bmatrix}
-k_{p\omega}\omega_f & 0 & 0 & 0 \\
0 & 0 & 0 & \frac{k_{qV}f_{q1}\omega_f}{f_{d1}m_{q1}-f_{q1}m_{d1}} \\
0 & 0 & 0 & \frac{k_{qV}f_{d1}\omega_f}{f_{q1}m_{d1}-f_{d1}m_{q1}} \\
0 & 0 & -k_{p\omega}\omega_f & 0 \\
0 & 0 & 0 & \frac{k_{qV}f_{q2}\omega_f}{f_{d2}m_{q2}-f_{q2}m_{d2}} \\
0 & 0 & 0 & \frac{k_{qV}f_{d2}\omega_f}{f_{q2}m_{d2}-f_{d2}m_{q2}}
\end{bmatrix}
\begin{bmatrix}
p_{N1} \\
q_{N1} \\
p_{N2} \\
q_{N2}
\end{bmatrix}
+
\begin{bmatrix}
\omega_f\omega_{ref} \\
-\frac{f_{q1}\omega_f}{f_{d1}m_{q1}-f_{q1}m_{d1}}V_{ref} \\
-\frac{f_{d1}\omega_f}{f_{q1}m_{d1}-f_{d1}m_{q1}}V_{ref} \\
\omega_f\omega_{ref} \\
-\frac{f_{q2}\omega_f}{f_{d2}m_{q2}-f_{q2}m_{d2}}V_{ref} \\
-\frac{f_{d2}\omega_f}{f_{q2}m_{d2}-f_{d2}m_{q2}}V_{ref}
\end{bmatrix}
\tag{A8}
$$

The Equation (A8) is rewritten in a matrix form

$$
\begin{bmatrix}
\frac{d\omega_1}{dt} \\
\frac{dv_{d1}}{dt} \\
\frac{dv_{q1}}{dt} \\
\frac{d\omega_2}{dt} \\
\frac{dv_{d2}}{dt} \\
\frac{dv_{q2}}{dt}
\end{bmatrix}
= A
\begin{bmatrix}
\omega_1 \\
v_{d1} \\
v_{q1} \\
\omega_2 \\
v_{d2} \\
v_{q2}
\end{bmatrix}
+ \mathbf{C}
\begin{bmatrix}
p_{N1} \\
q_{N1} \\
p_{N2} \\
q_{N2}
\end{bmatrix}
+ \mathbf{D}
\begin{bmatrix}
\omega_{ref} \\
V_{ref} \\
V_{ref} \\
\omega_{ref} \\
V_{ref} \\
V_{ref}
\end{bmatrix}
\tag{A9}
$$

where

$$
A =
\begin{bmatrix}
-\omega_f & 0 & 0 & 0 & 0 & 0 \\
\frac{m_{q1}}{f_{d1}m_{q1}-f_{q1}m_{d1}} & \frac{f_{q1}m_{d1}\omega_f}{f_{d1}m_{q1}-f_{q1}m_{d1}} & \frac{f_{q1}m_{q1}\omega_f}{f_{d1}m_{q1}-f_{q1}m_{d1}} & 0 & 0 & 0 \\
\frac{m_{d1}}{f_{q1}m_{d1}-f_{d1}m_{q1}} & \frac{f_{d1}m_{d1}\omega_f}{f_{q1}m_{d1}-f_{d1}m_{q1}} & \frac{f_{d1}m_{q1}\omega_f}{f_{q1}m_{d1}-f_{d1}m_{q1}} & 0 & 0 & 0 \\
0 & 0 & 0 & -\omega_f & 0 & 0 \\
0 & 0 & 0 & \frac{m_{q2}}{f_{d2}m_{q2}-f_{q2}m_{d2}} & \frac{f_{q2}m_{d2}\omega_f}{f_{d2}m_{q2}-f_{q2}m_{d2}} & \frac{f_{q2}m_{q2}\omega_f}{f_{d2}m_{q2}-f_{q2}m_{d2}} \\
0 & 0 & 0 & \frac{m_{d2}}{f_{q2}m_{d2}-f_{d2}m_{q2}} & \frac{f_{d2}m_{d2}\omega_f}{f_{q2}m_{d2}-f_{d2}m_{q2}} & \frac{f_{d2}m_{q2}\omega_f}{f_{q2}m_{d2}-f_{d2}m_{q2}}
\end{bmatrix}
\tag{A10}
$$

$$
C =
\begin{bmatrix}
-k_{p\omega}\omega_f & 0 & 0 & 0 \\
0 & 0 & 0 & \frac{k_{qV}f_{q1}\omega_f}{f_{d1}m_{q1}-f_{q1}m_{d1}} \\
0 & 0 & 0 & \frac{k_{qV}f_{d1}\omega_f}{f_{q1}m_{d1}-f_{d1}m_{q1}} \\
0 & 0 & -k_{p\omega}\omega_f & 0 \\
0 & 0 & 0 & \frac{k_{qV}f_{q2}\omega_f}{f_{d2}m_{q2}-f_{q2}m_{d2}} \\
0 & 0 & 0 & \frac{k_{qV}f_{d2}\omega_f}{f_{q2}m_{d2}-f_{d2}m_{q2}}
\end{bmatrix}
\tag{A11}
$$

$$
\mathbf{D} =
\begin{bmatrix}
\omega_f & 0 & 0 & 0 & 0 & 0 \\
0 & -\frac{f_{q1}\omega_f}{f_{d1}m_{q1}-f_{q1}m_{d1}} & 0 & 0 & 0 & 0 \\
0 & 0 & -\frac{f_{d1}\omega_f}{f_{q1}m_{d1}-f_{d1}m_{q1}} & 0 & 0 & 0 \\
0 & 0 & 0 & \omega_f & 0 & 0 \\
0 & 0 & 0 & 0 & -\frac{f_{q2}\omega_f}{f_{d2}m_{q2}-f_{q2}m_{d2}} & 0 \\
0 & 0 & 0 & 0 & 0 & -\frac{f_{d2}\omega_f}{f_{q2}m_{d2}-f_{d2}m_{q2}}
\end{bmatrix}
\tag{A12}
$$

Equation (A9) can be further rewritten as follows,

$$
\begin{bmatrix}
\frac{d\omega_1}{dt} \\
\frac{dv_{d1}}{dt} \\
\frac{dv_{q1}}{dt} \\
\frac{d\omega_2}{dt} \\
\frac{dv_{d2}}{dt} \\
\frac{dv_{q2}}{dt}
\end{bmatrix}
= A
\begin{bmatrix}
\omega_1 \\
v_{d1} \\
v_{q1} \\
\omega_2 \\
v_{d2} \\
v_{q2}
\end{bmatrix}
+ B
\begin{bmatrix}
\omega_{ref} \\
p_{N1} \\
q_{N1} \\
V_{ref} \\
p_{N2} \\
q_{N2}
\end{bmatrix}
\tag{A13}
$$

where

$$
B =
\begin{bmatrix}
\omega_f & -k_{p\omega}\omega_f & 0 & 0 & 0 & 0 \\
0 & 0 & 0 & -\frac{f_{q1}\omega_f}{f_{d1}m_{q1}-f_{q1}m_{d1}} & 0 & \frac{f_{q1}k_{qV}\omega_f}{f_{d1}m_{q1}-f_{q1}m_{d1}} \\
0 & 0 & 0 & \frac{f_{d1}\omega_f}{f_{d1}m_{q1}-f_{q1}m_{d1}} & 0 & -\frac{f_{d1}k_{qV}\omega_f}{f_{d1}m_{q1}-f_{q1}m_{d1}} \\
\omega_f & 0 & 0 & 0 & k_{p\omega}\omega_f & 0 \\
0 & 0 & 0 & -\frac{f_{q2}\omega_f}{f_{d2}m_{q2}-f_{q2}m_{d2}} & 0 & -\frac{f_{q2}k_{qV}\omega_f}{f_{d2}m_{q2}-f_{q2}m_{d2}} \\
0 & 0 & 0 & \frac{f_{d2}\omega_f}{f_{d2}m_{q2}-f_{q2}m_{d2}} & 0 & -\frac{f_{d2}k_{qV}\omega_f}{f_{d2}m_{q2}-f_{q2}m_{d2}}
\end{bmatrix}
\tag{A14}
$$

Around the equilibrium point $(i_{ed1}, i_{ed2}, i_{eq1}, i_{eq1})$, the power Equation (8) can be transformed into

$$
\begin{bmatrix}
p_1 \\
q_1 \\
p_2 \\
q_2
\end{bmatrix}
=
\begin{bmatrix}
i_{ed1} & i_{eq1} & 0 & 0 \\
-i_{eq1} & -i_{ed1} & 0 & 0 \\
0 & 0 & i_{ed2} & i_{eq2} \\
0 & 0 & -i_{eq2} & i_{ed2}
\end{bmatrix}
\begin{bmatrix}
v_{d1} \\
v_{q1} \\
v_{d2} \\
v_{q2}
\end{bmatrix}
\tag{A15}
$$

Based on the definition of network-based control shown in Equations (A1), (A2), and (A15), there is

$$
\begin{bmatrix}
\omega_{ref} \\
p_{N1} \\
q_{N1} \\
V_{ref} \\
p_{N2} \\
q_{N2}
\end{bmatrix}
=
\begin{bmatrix}
1 & 0 & 0 & 0 & 0 & 0 \\
0 & 1-m & 0 & 0 & m & 0 \\
0 & 0 & 1-n & 0 & 0 & n \\
0 & 0 & 0 & 1 & 0 & 0 \\
0 & m & 0 & 0 & 1-m & 0 \\
0 & 0 & n & 0 & 0 & 1-n
\end{bmatrix}
\begin{bmatrix}
\omega_{ref} \\
p_1 \\
q_1 \\
V_{ref} \\
p_2 \\
q_2
\end{bmatrix}
= V
\begin{bmatrix}
\omega_{ref} \\
p_1 \\
q_1 \\
V_{ref} \\
p_2 \\
q_2
\end{bmatrix}
\tag{A16}
$$

and

$$
\begin{bmatrix}
\omega_{ref} \\
p_1 \\
q_1 \\
V_{ref} \\
p_2 \\
q_2
\end{bmatrix}
=
\begin{bmatrix}
1 & k_{p\omega}(1-m)i_{ed1} & k_{p\omega}(1-m)i_{eq1} & 0 & k_{p\omega}(1-m)i_{ed2} & k_{p\omega}(1-m)i_{eq2} \\
0 & i_{ed1} & i_{eq1} & 0 & 0 & 0 \\
0 & -i_{eq1} & i_{ed1} & 0 & 0 & 0 \\
0 & 1+k_{qV}(1-n)i_{eq1} & k_{qV}(1-n)i_{ed1} & 0 & -k_{qV}ni_{eq2} & k_{qV}ni_{ed2} \\
0 & 0 & 0 & 0 & i_{ed2} & i_{eq2} \\
0 & 0 & 0 & 0 & -i_{eq2} & i_{ed2}
\end{bmatrix}
\begin{bmatrix}
\omega_1 \\
v_{d1} \\
v_{q1} \\
\omega_2 \\
v_{d2} \\
v_{q2}
\end{bmatrix}
= T
\begin{bmatrix}
\omega_1 \\
v_{d1} \\
v_{q1} \\
\omega_2 \\
v_{d2} \\
v_{q2}
\end{bmatrix}
\tag{A17}
$$

Then Equation (A16) can be rewritten by

$$
\begin{bmatrix}
\omega_{ref} \\
p_{N1} \\
q_{N1} \\
V_{ref} \\
p_{N2} \\
q_{N2}
\end{bmatrix}
= VT
\begin{bmatrix}
\omega_1 \\
v_{d1} \\
v_{q1} \\
\omega_2 \\
v_{d2} \\
v_{q2}
\end{bmatrix}
= K
\begin{bmatrix}
\omega_1 \\
v_{d1} \\
v_{q1} \\
\omega_2 \\
v_{d2} \\
v_{q2}
\end{bmatrix}
\tag{A18}
$$

Overall, the models of network-based control system, Equation (A13) and (A18), in autonomous mode are rewritten by

$$\dot{x}(t) = Ax(t) + Bu(t)$$
$$u(t) = Kx(t) \tag{A19}$$

where $x = \begin{bmatrix} \omega_1 & v_{d1} & v_{q1} & \omega_2 & v_{d2} & v_{q2} \end{bmatrix}^{\mathrm{T}}$ and $u = \begin{bmatrix} \omega_{ref} & p_{N1} & q_{N1} & V_{ref} & q_{N1} & q_{N2} \end{bmatrix}^{\mathrm{T}}$, the system matrix A is addressed in Equation (A10) and the system matrix B is written in Equation (A14) respectively. The feedback control matrix *K* is shown in Equation (A18). It is certain that the model can be similarly extended in master-slave mode and a situation with more than two paralleled inverters.

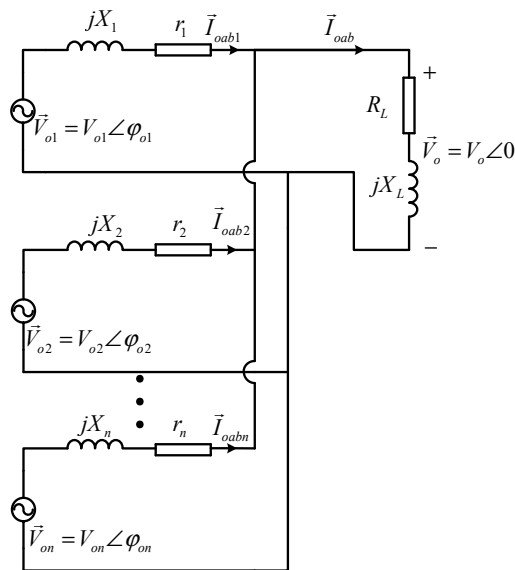

**Figure A1.** Simplified circuit of AB-phase for paralleled inverter system.

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
