# Peer review of "Analysis of Three-Phase Inverter Parallel Operation with Network-Based Control Having Strong Robustness and Wide Time-Scale Compatibility in Droop-Controlled AC Microgrid"

_electronics, doi:10.3390/electronics9020376_

Round 1

Reviewer 1 Report

The paper proposed a network-based droop control for parallel inverters. While there are many works of literature have been studied for the same object, the authors should clarify the contribution of the paper on both theory and experiment comparing with the other works. There are some issues as below need to be clarified to evaluate the paper. 

-The control block diagram of the proposed network-based control method should be presented to compare with the conventional method presented by fig. 4.

-The colour scale of fig. 8 needs to be explained.

-The contribution of the paper presented in section 4 should be presented more explicitly such as by in the form of a theorem.

-The network used for experimental assess should include at least 3 inverters to ensure that the proposed method is usable for the general case of the parallel inverters.

-A figure describes the environment of the experiment should be provided.

-How the proposed can be used with the inverters, which have different rated powers and different time delay levels?

-How the proposed system be operated when trouble occurs with the communication network?

-How about the THD of currents presented in section 5.3?

Reviewer 2 Report

Dear authors,

The paper proposes an interesting analysis of three-phase inverter parallel operation in AC microgrids. The paper extension is quite large but the appendix can help readers to understand the formulation.

In my modest point of view, there are several general questions that need more clarification:

1. What are the requirements of the network from the point of view of the latency and real-time response? In lines 340-341 you include a comment explaining that the system has a strong robustness towards time-delay but. You summarize this question in fig. 8 but a more detailed explanation is needed.

2. How robust is the proposed approach if network experiences communication problems?

From a more specific point of view:

Line 155. The first Zn is a complex magnitude. Please update the representation.
Line 167. "angel" should be "angle"
Line 176 and 343. Please, review table numeration. The first two tables are numbered as 1.

Line 224. Eq. (9) right term Qn has an extra prime

Line 294, 295 and 296. Please, review notation in equations (11), (12) and (13). The mutiplication/time symbol is a dot while in previous equations you don't use any symbol.

Line 421. Fig. 11. Please, review the input and output power flows in the inverter #2. Does the input to the Inverter #2 should be Pcom1 and Qcom1 and the output Pcal2 and Qcal2?

Line 474. The fonts of the legends included in figures 12 and 14 is too small. Please consider to increase it in order to improve the readability.

Line 619. Please review the numeration of the equations included in the appendix. There are two equations numbered as A18.

Thank you very much for your contribution

Best Regards

Round 2

Reviewer 2 Report

Dear authors,

I really appreciate so much your effort improving the paper.

Best Regards